# SELF-DISTILLATION FOR FURTHER PRE-TRAINING OF TRANSFORMERS

**Seanie Lee**[1†]   **Minki Kang**[1,2]   **Juho Lee**[1,2]   **Sung Ju Hwang**[1]   **Kenji Kawaguchi**[3*]

KAIST[1], AITRICS[2], National University of Singapore[3]
{lsnfamily02, zzxc1133, juholee, sjhwang82}@kaist.ac.kr
kenji@comp.nus.edu.sg

## ABSTRACT

The application of pre-training large transformer models on massive amounts of unlabeled data and fine-tuning them on labeled datasets for diverse downstream tasks has demonstrated remarkable success in various vision and natural language processing tasks. However, the direct fine-tuning approach may result in suboptimal performance if there exists a significant discrepancy between the pre-training and fine-tuning domains. To address this issue, some previous studies have proposed further pre-training strategies to continue pre-training the model on the target unlabeled dataset before fine-tuning. However, these strategies are limited to language models and may result in overfitting when applied to Vision Transformers. To overcome this limitation, we present a novel approach of self-distillation as a regularization method for the further pre-training stage. Our method first further pre-trains the initial pre-trained model on the target unlabeled data, and then uses it as a teacher for self-distillation. Then we take the same initial pre-trained model as a student, and enforce its hidden representations to be close to those of the teacher while optimizing the student with a masked auto-encoding objective. Our experiments demonstrate the superiority of self-distillation over relevant baselines on various benchmark datasets for image and text classification tasks. Furthermore, we provide a theoretical analysis of our proposed method using a simplified model to shed light on how self-distillation for further pre-training can potentially enhance the performance of downstream tasks.

## 1 INTRODUCTION

Pre-trained transformer models (Devlin et al., 2019; Brown et al., 2020; Liu et al., 2019; He et al., 2022) have been effective on various vision and natural language processing tasks. The pre-trained models learn general representation from a large volume of unlabeled data so that they generalize well to various downstream tasks when they are fine-tuned on each task with a labeled dataset. However, in many of real-world applications, it requires a considerable amount of effort to adapt the pre-trained model to a specific downstream task domain since there exists a significant distributional discrepancy between data for the pre-training and fine-tuning stage. Moreover, it is difficult to collect a large amount of labeled data for such specific domains, which renders adaptation of the pre-trained model to downstream tasks more challenging.

Several works have proposed to tackle the problem of adapting pre-trained models to a specific domain. A prevalent approach for adaptation of the pre-trained model is *further pre-training* where we continue to update the parameters of the pre-trained model on additionally curated domain-specific unlabeled data with self-supervision (Beltagy et al., 2019; Lee et al., 2020), before fine-tuning it on the target labeled data as depicted in Figure 2b. Gururangan et al. (2020) also show that further pre-training only with the target unlabeled data is still effective without any extra data. However, most of the existing further pre-training approaches have focused on language models, and we find that the further pre-training

Figure 1: Acc. with varying the number of further pre-training steps.

---

*Corresponding Author    †The work was done while the author was an intern at NUS.

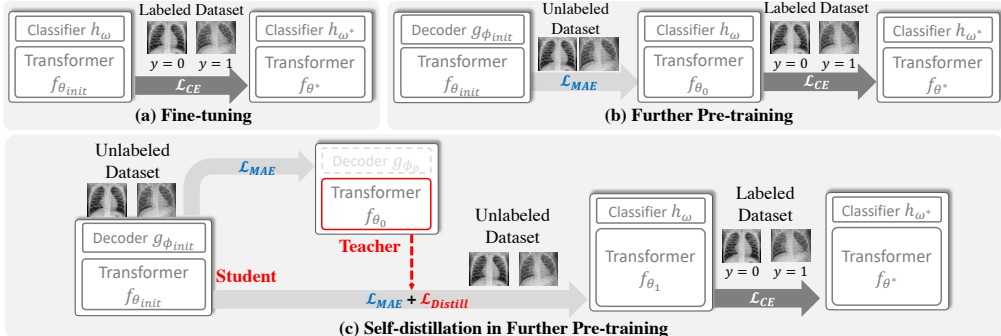

Figure 2: **Concepts.** Comparison between methods adapting pre-trained transformers to the target domain. **(a) Fine-tuning** without any further pre-training. **(b) Further pre-training** and fine-tuning. **(c) Self-distillation** in further pre-training and fine-tuning.

strategy is not effective for Vision Transformer (ViT) (Dosovitskiy et al., 2021). As shown in Figure 1, ViT is vulnerable to overfitting and does not generalize well to downstream tasks as when we continue to pre-train it on the target unlabeled data.

Several regularization methods (Chen et al., 2020a; Gouk et al., 2021; Aghajanyan et al., 2021) have proposed to tackle the overfitting issue of large pre-trained models, however, they do not consider the adaptation process such as further pre-training. Instead, they enforce the distance between the final fine-tuned weight and the pre-trained weight to be small to promote the transfer of the knowledge acquired from pre-training to downstream tasks for better generalization. However, these regularizations hinder the adaptation of pre-trained models to downstream tasks especially when there is a significant distributional shift between the pre-trained data and target data. It eventually results in worse generalization than the simple fine-tuning strategy.

To tackle these limitations, we propose *self-distillation* as a regularization for further pre-training on a target unlabeled dataset so that we can effectively adapt pre-trained models to the downstream task of various domains with a limited amount of labeled data. For self-supervision, we focus on masked auto-encoding for pre-training since it does not depend on any data augmentations, compared to other self-supervised learning methods (Chen et al., 2020b; He et al., 2020; Grill et al., 2020; Zbontar et al., 2021; Chen & He, 2021; Caron et al., 2021) which require data augmentations to construct positive pairs for self-supervised learning objective such as contrastive learning. This is especially useful when it is hard to define meaningful data augmentations for a target domain.

Specifically, we take the pre-trained model with an encoder $f_{\theta_{\text{init}}}$ and a decoder $g_{\phi_{\text{init}}}$ which are pre-trained on a massive amount of unlabeled data from general domain, and continue to pre-train it with masked auto-encoding (MAE) (Devlin et al., 2019; He et al., 2022) objective on the target unlabeled data to obtain $f_{\theta_0}$ and $g_{\phi_0}$. After that, we set the encoder $f_{\theta_0}$ as a teacher for self-distillation. Then we take the copy of the pre-trained model $(f_{\theta_{\text{init}}}, g_{\phi_{\text{init}}})$ as a student, and match the representations of the student encoder and those of the teacher encoder while optimizing the student with the MAE on the target unlabeled data. Finally, we fine-tune the self-distilled student $f_{\theta_1}$ on the target labeled data for the downstream task. We illustrate the overview of our method in Figure 2c.

To verify the efficacy of our method, we empirically show that it significantly improves the generalization performance of a pre-trained ViT and language model RoBERTA (Liu et al., 2019), and outperforms the relevant baselines on various image and text classification datasets. Moreover, we theoretically analyze the proposed method with a simplified model to understand how self-distillation for further pre-training can potentially help improve the generalization performance on the target tasks after fine-tuning.

Our contribution is threefold:

- We propose self-distillation for further pre-training on the target unlabeled dataset, where we enforce representations of the student to be close to those of the further pre-trained teacher while training the student with masked-auto encoding objective.

- We theoretically analyze the proposed method with a simplified model to understand how self-distillation for further pre-training can potentially lead to better generalization performance of downstream tasks.

- We extensively validate our method on various image and text classification datasets with pre-trained transformers and show that ours outperforms the relevant baselines.

## 2 RELATED WORK

**Self-Distillation** Knowledge distillation is to transfer knowledge of teacher to student by minimizing a divergence between output of teacher and student (Hinton et al., 2014). When the parameterization of student and teacher is identical, we call it *self-distillation* as a special case of the knowledge distillation. Although there is no new information during self-distillation process, Furlanello et al. (2018) have shown that the student from self-distillation achieves better generalization performance than the teacher. A similar phenomenon has been consistently observed in other works (Yang et al., 2019; Ahn et al., 2019). Several works propose a self-distillation without a pre-trained teacher network (Sun et al., 2019; Zhang et al., 2019; 2022). They add auxiliary classifiers to intermediate layers and train the classifiers to minimize divergence between the output of the classifier of the last layer and that of the auxiliary classifiers. Mobahi et al. (2020) theoretically analyze how self-distillation induces regularization and reduces overfitting in Hilbert space. However, all of them focus on self-distillation for supervised learning. Instead, we empirically and theoretically show that self-distillation for further pre-training with self-supervision leads to better generalization of downstream tasks after fine-tuning the self-distilled model with target labeled data.

**Further Pre-training** Lee et al. (2020); Beltagy et al. (2019); Sun et al. (2020) have shown the success of continual pre-training language model on a large number of corpora collected from target domain and fine-tuning the model on target labeled dataset. However, it is computationally expensive to further pre-train the model on a large amount of unlabeled text data and it may not be feasible to collect such a large scale of unlabeled data on certain domains. Instead, Gururangan et al. (2020) devise a task-adaptive pre-training where we use only target unlabeled data for further pre-training language model before fine-tuning the model on the target labeled data. To improve the effectiveness of further pre-training, Kang et al. (2020); Ye et al. (2021) propose learning to mask input for masked auto-encoding with bilevel optimization, which requires a prohibitive computational cost. However, all of them solely focus on pre-trained language models and we empirically find that naive further pre-training is not effective for Vision Transformers.

**Regularization for Fine-tuning** There are several works proposing regularization for fine-tuning a pre-trained model. Chen et al. (2020a) propose to modify Adam (Kingma & Ba, 2015) optimizer, called RecAdam, which enforces the fine-tuned model close to the initial pre-trained model by minimizing $L_2$ distance between fine-tuned and initial pre-trained weight. Similarly, Gouk et al. (2021) project the fine-tuned weight for every gradient descent update such that it lies within the sphere centered on the initial pre-trained weights with the distance induced by the norm of maximum absolute row sums (MARS). Instead of explicitly minimizing the distance, motivated by trust region theory, Aghajanyan et al. (2021) propose to minimize symmetric KL-divergence between the model output of an original input and that of the input perturbed by Gaussian noise. However, all of them do not consider adaptation of pre-trained models to a specific target domain, which results in worse generalization performance of downstream tasks than a simple fine-tuning strategy.

## 3 METHOD

### 3.1 PRELIMINARIES

**Problem Statement** We assume that we are given $(\theta_{\texttt{init}}, \phi_{\texttt{init}})$ parameters of the neural network $g_{\phi_{\texttt{init}}} \circ f_{\theta_{\texttt{init}}}$ which is pre-trained on a large volume of unlabeled data with masked auto-encoding objective, where $f_{\theta_{\texttt{init}}}$ is an encoder which extracts hidden representation of an input and $g_{\phi_{\texttt{init}}}$ is an decoder reconstructing a masked input. Our goal is to fine-tune the pre-trained model $f_{\theta_{\texttt{init}}}$ with a randomly initialized task specific head $h_\omega$ on labeled dataset $\mathcal{D}^{\texttt{tr}} = \{(\mathbf{x}^{(i)}, y^{(i)})\}_{i=1}^n$ of a downstream classification task such that the model generalizes well to unseen test dataset $\mathcal{D}^{\texttt{test}}$. A typical approach to achieve this goal is empirical risk minimization as follows:

$$\underset{\theta, \omega}{\text{minimize}} \ \mathcal{L}_{\text{CE}}(\theta, \omega; \mathcal{D}^{\texttt{tr}}) \ \text{ via algorithm } \mathcal{A} \text{ as}$$
$$(\theta^*, \omega^*) = \mathcal{A}(\mathcal{L}_{\text{CE}}; \theta_{\texttt{init}}, \mathcal{D}^{\texttt{tr}}), \tag{1}$$

---

**Algorithm 1** Self-Distillation

**Require:** Unlabeled dataset $\mathcal{D}^u$, initialization $\theta_{\texttt{init}}, \phi_{\texttt{init}}$, learning rate $\alpha \in \mathbb{R}_{\geq 0}$, round of self-distillation $T' \in \mathbb{N}_+$, masking probability $\gamma \in (0, 1)$ and batch size $B$.
1: $\theta_0 \leftarrow \text{Further\_Pretrain}(\mathcal{D}^u, \theta_{\texttt{init}}, \phi_{\texttt{init}}, \alpha, \gamma, B)$
2: **for all** $t \leftarrow 1, \ldots, T'$ **do**
3:     Initialize $\theta_t \leftarrow \theta_{\texttt{init}}$ and $\phi_t \leftarrow \phi_{\texttt{init}}$
4:     **while** not converge **do**
5:        Sample a mini-batch $\{\mathbf{x}^{(j)}\}_{j=1}^B$ from $\mathcal{D}^u$
6:        **for all** $j \leftarrow 1, \ldots, B$ **do**
7:           Sample a mask $\mathbf{z}^{(j)} \sim p_{\gamma, K}(\mathbf{z})$
8:           $Z^{(j)} \leftarrow \sum_{k=1}^K z_k^{(j)}$
9:           Get a masked input $\hat{\mathbf{x}}^{(j)}$ with $\mathbf{z}^{(j)}$
10:          $\ell_j^1 \leftarrow -\sum_{k=1}^K \frac{z_k^{(j)}}{Z^{(j)}} \log p_{\theta_t, \phi_t}(x_k^{(j)}|\hat{\mathbf{x}}^{(j)})$
11:          $\ell_j^2 \leftarrow \left\| f_{\theta_t}(\mathbf{x}^{(j)}) - \texttt{StopGrad}(f_{\theta_0}(\mathbf{x}^{(j)})) \right\|_2^2$
12:        **end for**
13:        $\mathcal{L}_1 \leftarrow \frac{1}{B}\sum_{j=1}^B \ell_j^1, \mathcal{L}_2 \leftarrow \frac{1}{B}\sum_{j=1}^B \ell_j^2$
14:        $\theta_t \leftarrow \theta_t - \alpha \frac{\partial(\mathcal{L}_1 + \mathcal{L}_2)}{\partial\theta}|_{\theta=\theta_t}$
15:        $\phi_t \leftarrow \phi_t - \alpha \frac{\partial\mathcal{L}_1}{\partial\phi}|_{\phi=\phi_t}$
16:     **end while**
17:     $\theta_0 \leftarrow \theta_t$
18: **end for**
19: **return** $\theta_{T'}$

---

**Algorithm 2** Further_Pretrain

**Require:** Unlabeled dataset $\mathcal{D}^u$, initialization $\theta_{\texttt{init}}, \phi_{\texttt{init}}$, learning rate $\alpha \in \mathbb{R}_{\geq 0}$, masking probability $\gamma \in (0, 1)$, and batch size $B$.
1: Initialize $\theta_0 \leftarrow \theta_{\texttt{init}}$ and $\phi_0 \leftarrow \phi_{\texttt{init}}$
2: **while** not converge **do**
3:     Sample a mini-batch $\{\mathbf{x}^{(j)}\}_{j=1}^B$ from $\mathcal{D}^u$

4:     **for all** $j \leftarrow 1, \ldots, B$ **do**
5:        Sample a mask $\mathbf{z}^{(j)} \sim p_{\gamma, T}(\mathbf{z})$
6:        $Z^{(j)} \leftarrow \sum_{k=1}^K z_k^{(j)}$
7:        Get a masked input $\hat{\mathbf{x}}^{(j)}$ with $\mathbf{z}^{(j)}$
8:        $p_k \leftarrow p_{\theta_0, \phi_0}(x_k^{(j)}|\hat{\mathbf{x}}^{(j)})$
9:        $\ell_j^1 \leftarrow -\sum_{k=1}^K \frac{z_k^{(j)}}{Z^{(j)}} \log p_k$
10:     **end for**
11:     $\mathcal{L} \leftarrow \frac{1}{B}\sum_{j=1}^B \ell_j^1$
12:     $\theta_0 \leftarrow \theta_0 - \alpha \frac{\partial\mathcal{L}}{\partial\theta}|_{\theta=\theta_0}$
13:     $\phi_0 \leftarrow \phi_0 - \alpha \frac{\partial\mathcal{L}}{\partial\phi}|_{\phi=\phi_0}$
14: **end while**
15: **return** $\theta_0$

---

where $\mathcal{L}_{\texttt{CE}}$ is a cross-entropy loss and $\mathcal{A}$ denotes a stochastic gradient descent algorithm to minimize $\mathcal{L}_{\texttt{CE}}$ on the dataset $\mathcal{D}^{\texttt{tr}}$ with the initialization $\theta_{\texttt{init}}$.

**Further Pre-training**    However, the pre-trained model is prone to overfitting when it is fine-tuned on a small amount of domain-specific labeled data. Gururangan et al. (2020) have shown that further pre-training, where we continue to pre-train the model $g_{\phi_{\texttt{init}}} \circ f_{\theta_{\texttt{init}}}$ on the target unlabeled dataset $\mathcal{D}^u = \{\mathbf{x}^{(i)}\}_{i=1}^n$ and then fine-tune it on $\mathcal{D}^{\texttt{tr}}$, is effective for improving generalization performance when there is not enough domain-specific labeled data. Note that $\mathcal{D}^u$ is the exactly same as $\mathcal{D}^{\texttt{tr}}$ except that we remove the labels $y^{(i)}$. In this work, we focus on the masked auto-encoding (Devlin et al., 2019; He et al., 2022) as a pre-training objective function since its generality compared to other self-supervised methods (Chen et al., 2020b; He et al., 2020; Grill et al., 2020; He et al., 2020; Chen & He, 2021; Caron et al., 2021) which require well-defined data augmentations to construct positive pairs for self-supervised learning.

**Masked Auto-Encoding**    We briefly describe the masked auto-encoding objective (Liu et al., 2019; He et al., 2022) for a language model such as RoBERTA (Liu et al., 2019) and Vision Transformer (ViT) (Dosovitskiy et al., 2021). Let $\mathbf{x}^{(i)} = (x_1^{(i)}, \ldots, x_K^{(i)})$ be a sequence of patches for a image or tokens for a sentence with length $K$. Then we independently sample a binary mask from Bernoulli distribution with probability $\gamma$ for each $x_k^{(i)}$, denoted as $\mathbf{z}^{(i)} = (z_1^{(i)}, \ldots, z_K^{(i)})$. If $z_k^{(i)} = 1$, then $x_k^{(i)}$ is replaced with a special "mask" token. Otherwise, we use the same $x_k^{(i)}$ for a masked input. Let $\hat{\mathbf{x}}^{(i)} = (\hat{x}_1^{(i)}, \ldots, \hat{x}_K^{(i)})$ be a masked input and let $f_\theta, g_\phi$ be an encoder and decoder, respectively. Then the final objective for masked auto-encoding is defined as follows:

$$\mathcal{L}_{\texttt{MAE}}(\theta, \phi; \mathcal{D}^u) = \frac{1}{n}\sum_{i=1}^n \mathbb{E}_{\mathbf{z}^{(i)} \sim p_{\gamma, T}(\mathbf{z})} \left[ -\sum_{k=1}^K \frac{z_k^{(i)}}{Z^{(i)}} \cdot \log p_{\theta, \phi}(x_k^{(i)}|\hat{\mathbf{x}}^{(i)}) \right], \quad Z^{(i)} = \sum_{k=1}^K z_k^{(i)}, \quad (2)$$

where $p_{\gamma, K}(\mathbf{z})$ denotes a Binomial distribution with its parameters $\gamma$ for probability that $z_k = 1$ and $K$ for the number of trials. Note that the negative log-likelihood is instantiated as cross-entropy loss for language models or mean square error for Vision Transformers. See Appendix C for more detail.

## 3.2   SELF-DISTILLATION FOR FURTHER PRE-TRAINING

Although further pre-training strategy has been effective on text domain (Gururangan et al., 2020; Lee et al., 2020; Sun et al., 2020), we empirically find that ViT with further pre-training overfits

the target unlabeled data and does not generalize well to downstream image classification tasks. In order to tackle the issue, we propose self-distillation as a regularization for further pre-training. Specifically, given a pre-trained model $g_{\phi_{\text{init}}} \circ f_{\theta_{\text{init}}}$, we first continue to train the model on the target unlabeled data $\mathcal{D}^u$ with the masked auto-encoding objective as described in equation 2 to obtain the encoder $f_{\theta_0}$ and decoder $g_{\phi_0}$. We discard the decoder and consider the encoder $f_{\theta_0}$ as a teacher for self-distillation. Then we take the copy of the pre-trained initial network $g_{\phi_{\text{init}}} \circ f_{\theta_{\text{init}}}$ as a student and further pre-train the student with masked auto-encoding objective but enforce hidden representation of the encoder of the student $f_{\theta_{\text{init}}}$ to be close to that of the teacher $f_{\theta_0}$ as follows:

$$(\theta_1, \phi_1) \in \arg\min_{\theta, \phi} \left( \mathcal{L}_{\text{MAE}}(\theta, \phi; \mathcal{D}^u) + \mathcal{L}_{\text{Distill}}(\theta; \theta_0, \mathcal{D}^u) \right)$$

$$\mathcal{L}_{\text{Distill}}\left(\theta; \theta_0, \mathcal{D}^u\right) = \frac{1}{n} \sum_{i=1}^{n} \left\| f_\theta(\mathbf{x}^{(i)}) - \text{StopGrad}\left( f_{\theta_0}(\mathbf{x}^{(i)}) \right) \right\|_2^2 \tag{3}$$

where $\theta$ and $\phi$ are initialized with the pre-trained parameters $\theta_{\text{init}}$ and $\phi_{\text{init}}$, respectively and StopGrad denotes the stop-gradient operation which does not back-propagate through the input. As described in Algorithm 1, we can repeat this process to perform multiple rounds of self-distillation ($T' > 1$) where the student of the previous round becomes a teacher and a new student is initialized with the pre-trained weights $\theta_{\text{init}}$ and $\phi_{\text{init}}$ for the next round. We empirically find that the first round of self-distillation plays the most significant role in improving the final generalization performance of downstream tasks. Furthermore, theoretical analysis shows that the first round of self-distillation has the largest impact on regularization. Thus, we perform a single round of self-distillation for computational efficiency. After self-distillation, we discard the decoder $g_{\phi_1}$ and fine-tune the encoder of the student $f_{\theta_1}$ along with a randomly initialized task-specific head $h_\omega$ by minimizing $\mathcal{L}_{\text{CE}}(\theta, \omega, ; \mathcal{D}^{\text{tr}})$ with the initialization $\theta_1$ as described in equation 1.

# 4 THEORETICAL ANALYSIS

In this section, we analyze how self-distillation affects the final model after fine-tuning in terms of generalization and regularization. This section proves a generalization bound on the supervised loss for our method and shows that the generalization bound strictly decreases as the number of self-distillation increases. Moreover, we show that self-distillation acts as a regularizer on the distance between the initial weight before further pre-training and the final weight after fine-tuning. The regularization effect is shown to have the largest impact in the first round of self-distillation, which suggests that the first round of self-distillation plays a more significant role in the final performance when compared to the other rounds.

We consider the dynamics of the weight vector $w_{t,\tau}$ over time $\tau$ of fine-tuning after $t$ rounds of self-distillation, where $w_{0,0}$ is the result of further pre-training, and $w_{t,0} \in \text{minimize}_w L_t(w)$ is the result of the self-distillation of $t$ rounds with $L_t(w) = \frac{1}{n} \sum_{i=1}^{n} \|f(x_i, w) - f(x_i, w_{t-1,0})\|_2^2 + \lambda \|w\|_2^2$ for some $\lambda > 0$. After $t$ rounds of self-distillation, we consider the dynamics over fine-tuning time $\tau$ via gradient flow (Saxe et al., 2014; Kawaguchi, 2021): $\frac{dw_{t,\tau}}{d\tau} = -\nabla\mathcal{L}(w_{t,\tau})$, with the initialization $w_{t,0}$ obtained by the self-distillation where $\mathcal{L}(w) = \frac{1}{2} \sum_{i=1}^{n} \ell(w, x_i, y_i)$ with $\ell(w, x, y) = \|f(x, w) - y\|_2^2$ and $y \in \mathbb{R}^p$. Here, the self-distillation and fine-tuning share a same training dataset $s = \{(x_i, y_i)\}_{i=1}^n$. In this section, to obtain theoretical insights, we consider the regime of $d > n$ and a simple abstract model, $f(x, w) = W\varphi(x) \in \mathbb{R}^p$, with some nonlinear map $\varphi$ and the weight matrix $W \in \mathbb{R}^{p \times d}$ where $w = \text{vec}[W^\top] \in \mathbb{R}^{dp}$ and $\varphi(x) \in \mathbb{R}^d$. Here, $\text{vec}[W^\top]$ is a vectorization of the matrix $W^\top$. Let us fix the fine-tuning time length $T$ as $1 < \tau \leq T < \infty$. Since $d > n$, there are infinitely many solutions to the problem of minimizing $\mathcal{L}(w)$. Thus, each of the finite length $T$ and the over-parameterization $d > n$ implies that the initialization $w_{t,0}$ at the fine-tuning phase via self-distillation plays an important role.

Let $\delta > 0$ and $t \in \mathbb{N}_0$. We then define $\mathcal{F}_t = \{\mathcal{A}_t(s) : s \in \mathcal{S}\}$, where $\mathcal{S}$ is a set of all training datasets of size $n$ such that with probability at least $1 - \delta$, the training dataset $s$ is in $\mathcal{S}$. For each training dataset $s \in \mathcal{S}$, $\mathcal{A}_t(s) = w_{t,T}$ is the final weight vector of the model after $t$ rounds of self-distillation and $T$ time of fine-tuning. Let us define the matrix $\Phi \in \mathbb{R}^{d \times n}$ by $\Phi_{ij} = \varphi(x_j)_i$. We assume that $\Phi$ is of full rank; i.e., $\text{rank}(\Phi) = n$ since $d \geq n$. This is typically satisfied because if $\text{rank}(\Phi) < n$, there is some redundancy in the rows of the matrix $\Phi$. Denote by $[I_p \otimes \Phi] \in \mathbb{R}^{dp \times np}$ the Kronecker product of the identity matrix $I_p \in \mathbb{R}^{p \times p}$ and the matrix $\Phi$. We write its singular

value decomposition by $[I_p \otimes \Phi] = U \Sigma V^\top$ where $U = [u_1, u_2 \ldots, u_{dp}] \in \mathbb{R}^{dp \times dp}$ contains the left-singular vectors $u_i \in \mathbb{R}^{dp}$ for $i \in \{1, \ldots, dp\}$ and $\Sigma \in \mathbb{R}^{dp \times np}$ is a rectangular diagonal matrix with $\Sigma_{ii} = \sigma_i \in \mathbb{R}_{\geq 0}$ for $i \in \{1, \ldots, np\}$ and $\sigma_1 \geq \sigma_2 \geq \cdots \geq \sigma_{np} \geq 0$. Define $M$ to be an upper bound on the loss as $\ell(w, x, y) \leq M$. Define $R$ to be an upper bound on the expected norm of the features as $\mathbb{E}_x \|\varphi(x)\|_2 \leq R$. We assume that $w_{0,0} \neq 0$; if $w_{0,0} = 0$, then the target function in the self-distillation phase is always zero as $f(x_i, w_{0,0}) = 0$ for all $i$, which is unlikely the case in practice. We define $w_{\text{init}} \in \mathbb{R}^{dp}$ to be the weight before further pre-training and define $Y = \text{vec}[[y_1, \ldots, y_n]^\top] \in \mathbb{R}^{np}$.

The following theorem shows that the generalization bound on the supervised loss $\ell(w_{t,T}, x, y)$ of the fine-tuning phase strictly decreases as we increase the number $t$ of self-distillation rounds in the further pre-training phase:

**Theorem 1.** *There exists a constant $c$ (that only depends on $M$) such that with probability at least $1 - \delta$, the following holds:*

$$\mathbb{E}_{x,y}[\ell(w_{t,T}, x, y)] \leq \frac{1}{n} \sum_{i=1}^{n} \ell(w_{t,T}, x_i, y_i) + \zeta(t)\sqrt{\frac{4c^2 R^2 p}{n}} + M\sqrt{\frac{\ln(2/\delta)}{2n}}, \qquad (4)$$

*where the function $\zeta(t)$ is strictly decreasing in $t \in \mathbb{N}_0$.*

The proofs of all results in this section are presented in Appendix A. Moreover, the following theorem shows that the tight upper bound on the distance between the initial weight $w_{\text{init}}$ and the final weight $w_{t,T}$ after $T$ steps of fine-tuning (i.e., $\|w_{\text{init}} - w_{t,T}\|_2$) strictly decreases as the number $t$ of self-distillation rounds increases:

**Theorem 2.** *There is a function $\psi : \mathbb{N}_0 \to \mathbb{R}_{\geq 0}$ such that (1) $\|w_{\text{init}} - w_{t,T}\|_2 = \psi(t)$ for some $w_{\text{init}} \in \mathbb{R}^{dp}$, (2) $\|w_{\text{init}} - w_{t,T}\|_2 \leq \psi(t)$ for all $w_{\text{init}} \in \mathbb{R}^{dp}$, (3) the function $\psi(t)$ is strictly decreasing in $t \in \mathbb{N}_0$, (4) the function $\psi(t)$ can be decomposed to $\psi(t) = \sqrt{G_1 + \psi_1(t) + \mathbb{1}\{t = 0\}\mathcal{B}} + G_2$ with constants $G_1, G_2 \geq 0$ in $t$ where $\psi_1(t)$ is strictly decreasing in $t \in \mathbb{N}_0$ and $\mathcal{B} = \sum_{i=np+1}^{dp}(u_i^\top w_{0,0})^2 \geq 0$.*

Theorem 2 shows that the self-distillation acts as a regularizer on the distance between the initial weight $w_{\text{init}}$ and the final weight $w_{t,T}$. Since the Rademacher complexity of a set of vectors is invariant to a shift by a constant vector, this distance has been shown to control the generalization bound in previous papers in various models and settings, including deep neural networks (Bartlett & Mendelson, 2002; Bartlett et al., 2017; Nagarajan & Kolter, 2019). This suggests that self-distillation helps generalization via a regularization effect on the distance. Moreover, the first round of self-distillation is expected to have the largest impact based on Theorem 2 since Theorem 2 shows that we can completely remove the unnecessary component $\mathcal{B}$ of $w_{0,0}$ in the first round of self-distillation. We have verified these theoretical predictions in the experiments where we show the correlation between the improvement via self-distillation and the distance that appeared in the generalization bound in the previous paper (Nagarajan & Kolter, 2019).

## 5 EXPERIMENT

**Dataset** For image classification problem, we use six datasets — FGVC Aircraft (Aircraft) (Maji et al., 2013), Caltech UCSD Birds 200 (CUB) (Wah et al., 2011), Chest X-ray (Kermany et al., 2018), Describable Textures Dataset (DTD) (Cimpoi et al., 2014), Stanford Dogs (Khosla et al., 2011), and Oxford 102 Flower (Nilsback & Zisserman, 2008). For text classification problem, we use four datasets — Chemprot (Kringelum et al., 2016), ACL-ARC (Jurgens et al., 2018), SCIERC (Luan et al., 2018), and Twitter-Emotion (Mohammad et al., 2018). Please see Appendix D for more detail.

**Implementation Detail** For the image classification problem, we use Vision Transformer pre-trained on unlabeled ImageNet dataset with masked auto-encoding (He et al., 2022) and fine-tune it on the downstream task with AdamW optimizer (Loshchilov & Hutter, 2019) for 10,000 steps with batch size 32. Regarding further pre-training and self-distillation, we continue to pre-train the model for 20,000 steps with batch size 64. We evaluate the Vision Transformers with accuracy. For text classification, following the experimental setup from Gururangan et al. (2020), we use RoBERTA (Liu et al., 2019) as a backbone network and fine-tune it on the target labeled dataset

Table 1: Average and standard deviation of accuracy with 5 runs for image classification datasets.

| Method | Aircraft | CUB | Chest X-ray | DTD | Dogs | Flower |
|---|---|---|---|---|---|---|
| Fine-tuning | $72.33 \pm 1.13$ | $55.55 \pm 0.54$ | $77.15 \pm 0.52$ | $67.56 \pm 0.52$ | $62.53 \pm 0.57$ | $88.78 \pm 0.65$ |
| RecAdam | $70.76 \pm 1.25$ | $55.22 \pm 1.29$ | $77.29 \pm 1.32$ | $67.59 \pm 1.03$ | $61.65 \pm 0.92$ | $88.97 \pm 0.44$ |
| MARS | $72.74 \pm 0.57$ | $55.35 \pm 0.73$ | $77.28 \pm 1.80$ | $66.79 \pm 0.90$ | $62.24 \pm 0.96$ | $87.93 \pm 1.21$ |
| R3F | $72.95 \pm 0.46$ | $55.91 \pm 0.79$ | $76.86 \pm 0.97$ | $65.32 \pm 1.25$ | $62.15 \pm 0.48$ | $88.92 \pm 0.78$ |
| Further Pre-training | $73.38 \pm 0.64$ | $55.72 \pm 0.46$ | $77.79 \pm 2.06$ | $65.55 \pm 1.12$ | $62.34 \pm 0.39$ | $88.63 \pm 0.35$ |
| **Self-Distillation** | $\mathbf{74.37 \pm 0.43}$ | $\mathbf{58.06 \pm 0.90}$ | $\mathbf{79.68 \pm 1.05}$ | $\mathbf{68.51 \pm 0.51}$ | $\mathbf{63.55 \pm 0.39}$ | $\mathbf{90.28 \pm 0.44}$ |

Table 2: Average and standard deviation of F1 score with 5 runs for text classification datasets.

| Method | SCIERC | ACL-ARC | Chemprot | Twitter-Emotion |
|---|---|---|---|---|
| Fine-tuning | $76.63 \pm 2.06$ | $64.09 \pm 4.13$ | $80.59 \pm 1.15$ | $77.61 \pm 0.83$ |
| RecAdam | $79.45 \pm 1.92$ | $59.70 \pm 2.68$ | $\mathbf{82.73 \pm 0.28}$ | $78.26 \pm 0.88$ |
| MARS | $74.69 \pm 1.25$ | $53.57 \pm 6.64$ | $80.18 \pm 0.92$ | $77.48 \pm 1.69$ |
| R3F | $75.61 \pm 2.49$ | $60.13 \pm 2.55$ | $79.25 \pm 2.16$ | $77.79 \pm 0.81$ |
| Further Pre-training | $80.32 \pm 1.25$ | $69.73 \pm 2.40$ | $82.33 \pm 0.46$ | $78.71 \pm 0.40$ |
| **Self-Distillation** | $\mathbf{81.79 \pm 0.75}$ | $\mathbf{73.17 \pm 2.19}$ | $\mathbf{82.87 \pm 0.35}$ | $\mathbf{79.77 \pm 0.79}$ |

with AdamW optimizer for 10 epochs with batch size 32. In terms of further pre-training and self-distillation, we further pre-train RoBERTA for 100 epochs with batch size 128. We evaluate the models with macro F1 for SCIERC, ACL-ARC, and Twitter-Emotion dataset, and micro F1 for Chemprot dataset.

**Baselines** We compare our method against the following baselines targeting for fine-tuning pre-trained models. All the models are initialized with the pre-trained weights $\theta_{\texttt{init}}$ and $\phi_{\texttt{init}}$.

1. **Fine-tuning**: The model fine-tuned on target labeled dataset $\mathcal{D}^{\texttt{tr}}$ without any further pre-training or regularization except dropout and weight decay.

2. **RecAdam** (Chen et al., 2020a): The model trained with RecAdam optimizer which is a variant of Adam optimizer (Kingma & Ba, 2015) and additionally penalizes $L_2$ distance between the fine-tuned and the initial pre-trained weight.

3. **MARS** (Gouk et al., 2021): The model trained to minimize the cross-entropy loss along with the regularization projecting the fine-tuned weight to lie within a sphere centered on the initial pre-trained weights. For each layer, the distance induced by Maximum Absolute Row Sum (MARS) matrix norm $(\max_j \sum_{i=1} |W_{j,i} - U_{j,i}|)$ is used for the regularization.

4. **R3F** (Aghajanyan et al., 2021): The model trained to minimize the cross-entropy loss as well as symmetric KL-divergence between softmax output of the original input and that of the input perturbed by Gaussian noise.

5. **Further Pre-training** (Gururangan et al., 2020): Task adaptive pre-training where we further pre-train the model on the unlabeled target dataset $\mathcal{D}^u$ with masked auto-encoding objective and fine-tune it on the target labeled dataset $\mathcal{D}^{\texttt{tr}}$.

6. **Self-Distillation**: This is our model which is further pre-trained on unlabeled target dataset $\mathcal{D}^u$ with equation 3 and fine-tuned on the target labeled dataset $\mathcal{D}^{\texttt{tr}}$.

## 5.1 MAIN RESULTS

As shown in Table 1, self-distillation consistently outperforms all the regularization methods and the further pre-training method on image datasets. Notably, our method significantly improves the performance of the Chest X-ray dataset consisting of grey-scaled images for diagnosis of pneumonia. In addition, self-distillation effectively tackles the Flower dataset which contains only 2,040 labeled examples. In contrast, the other baselines do not show consistent improvement across all the image datasets. For instance, further pre-training is effective for the Aircraft dataset, but significantly degrades the test accuracy on the DTD dataset. Regularization methods such as RecAdam, MARS, and R3F barely improve generalization performance on most datasets or underperform the simple fine-tuning strategy on certain datasets. This empirical evidence supports that the regularizations enforcing the fine-tuned models close to the initial pre-trained weight are not effective for adapting a pre-trained model to the target datasets of specific domains.

Furthermore, as shown in Table 2, we provide additional experimental results for text classification tasks. Again, self-distillation significantly outperforms all of the baselines across all four datasets,

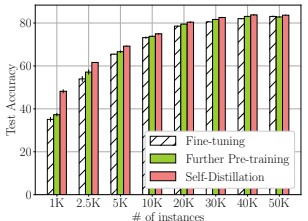

Figure 3: Accuracy with varying the number of training data.

Table 3: Ablation on CUB and SCIERC.

| Model | CUB | SCIERC |
|---|---|---|
| Full Model | **58.06 ± 0.90** | **81.79 ± 0.75** |
| w/o $\mathcal{L}_{\mathrm{MAE}}$ | 57.60 ± 0.81 | 80.66 ± 0.62 |
| w/o $\mathcal{L}_{\mathrm{Distill}}$ | 55.72 ± 0.46 | 80.32 ± 1.25 |
| Further Pre-train×2 | 53.41 ± 0.75 | 80.52 ± 0.98 |
| Prediction-Matching | 55.27 ± 1.07 | 81.09 ± 1.07 |
| Weight-Matching ($\ell_2$) | 53.70 ± 0.91 | 80.54 ± 1.21 |
| Weight-Matching (MARS) | 54.82 ± 0.60 | 80.95 ± 0.71 |

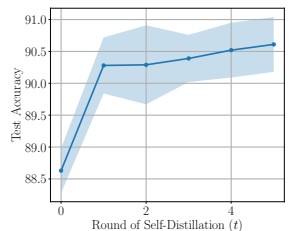

Figure 4: Test accuracy with varying self-distillation round.

except RecAdam in the Chemprot dataset. In contrast to the previous experiment, the further pre-training method improves the test F1 score of the simple fine-tuning method, yet it still underperforms our model. For regularization methods — RecAdam, MARS, and R3F, they do not achieve consistent improvement across all three datasets. RecAdam moderately improves the F1 score on the SCIERC and Chemprot dataset but significantly degrades the generalization performance on ACL-ARC dataset. Both MARS and R3F show poor performance on SCIERC and ACL-ARC datasets, and their performance slightly is worse than Fine-tuning method on the Chemprot dataset.

**Result for Low Resource Data** We further perform experiments to show how self-distillation effectively handles low resources of labeled data. Given a full CIFAR-100 dataset (Krizhevsky et al., 2009) which contains 50,000 training pairs of an image and corresponding label, we plot the test accuracy of each model by varying the number of training instances. Note that we also reduce the number of unlabeled images used for further pre-training or self-distillation. As shown in Figure 3, self-distillation consistently improves the generalization performance of both fine-tuning method and the model which is further pre-trained on the images from the CIFAR-100 dataset. Notably, the gain by self-distillation becomes larger when the models are trained with an extremely small number of instances. For example, self-distillation achieves 13% and 6% improvement of test accuracy compared to the model with simple fine-tuning when there are 1,000 and 2,500 labeled examples, respectively. These empirical results verify that self-distillation can effectively adapt the pre-trained model to the target dataset even if there are extremely small amounts of labeled data.

**Ablation Study** We perform ablation study to verify the effectiveness of each component of self-distillation. In Table 3, we show empirical results on both the CUB dataset and SCIERC data set while removing or replacing various components of self-distillation. Firstly, we remove masked auto-encoding objective $\mathcal{L}_{\mathrm{MAE}}$ and train the model with only distillation loss $\mathcal{L}_{\mathrm{Distill}}$ before fine-tuning. On image dataset CUB, it does not make a significant difference, however, removing the masked auto-encoding objective degrades the generalization performance of the language model on text classification dataset SCIERC. Alternatively, we remove the distillation loss $\mathcal{L}_{\mathrm{Distill}}$ in equation 3, which results in further pre-training method. Furthermore, we continue to pre-train the model for twice longer steps as the original further pre-training method, denoted as Further Pre-train×2, to show that higher test accuracy of self-distillation is not a consequence of longer pre-training. Both of the models significantly underperform self-distillation, which shows the effectiveness of the self-distillation loss. Lastly, we perform experiments for variants of distillation loss $\mathcal{L}_{\mathrm{Distill}}$ in equation 3. Instead of matching representation of teacher and student, we enforce the reconstruction of masked inputs by teacher and student to be consistent, i.e., $\mathrm{minimize}_{\theta,\phi} \|g_\phi \circ f_\theta(\hat{\mathbf{x}}) - g_{\phi_0} \circ f_{\theta_0}(\hat{\mathbf{x}})\|_2^2$ for ViT or $\mathrm{minimize}_{\theta,\phi} \sum_{t=1}^{T} D_{\mathrm{KL}}\left(p_{\theta_0,\phi_0}(x_t|\hat{\mathbf{x}}) \| p_{\theta,\phi}(x_t|\hat{\mathbf{x}})\right)$ for RoBERTA, denoted as Prediction-Matching. Furthermore, we replace the distillation loss with the one minimizing $L_2$ or MARS distance between the parameters of student and teacher, denoted as Weight-Matching. As shown in Table 3, all these variants are not effective compared to the one minimizing the distance between hidden representations of the student and teacher.

**Multi-Round of Self-Distillation** Lastly, we empirically show that the first round of self-distillation plays the most significant role in improving generalization performance. Specifically, we fine-tune each model after $t$ round of self-distillation and plot the test accuracy on Oxford 102 Flower dataset, where 0 round of self-distillation ($t = 0$) denotes the model with further pre-training. As shown in Figure 4, the first round of self-distillation significantly improves the test accuracy of the model with further pre-training and the gain by self-distillation becomes marginal after the first round. Considering the extra computational cost and marginal improvement of multi-round self-distillation, we perform a single round of self-distillation for all the experiments.

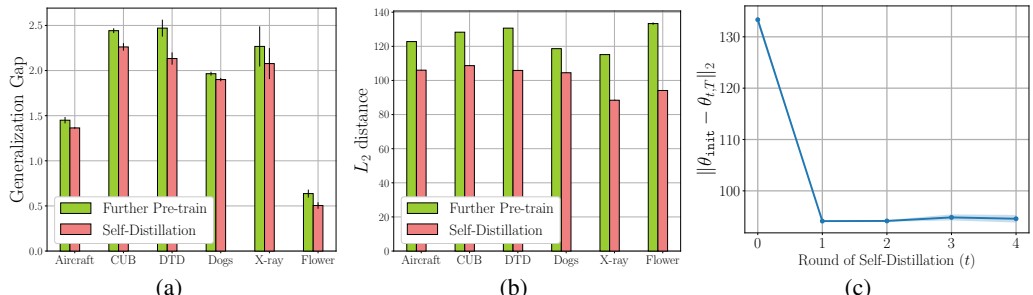

Figure 5: **(a) Generalization gap**: Difference between supervised test loss and training loss. **(b) Effect of self-distillation on distance:** Distance between the initial pre-trained weights and the final fine-tuned weights of further pre-training and self-distillation. **(c) Effect of multi-round self distillation:** Distance between the initial pre-trained weights and the final fine-tuned weights for each round of self-distillation $t \in \mathbb{N}_+$.

## 5.2 FURTHER ANALYSIS

In this subsection, we present numerical experiments to analyze why self-distillation can potentially help improve the generalization performance of downstream tasks compared to further pre-training and empirically show that Theorem 1 and 2 can be extended to deep neural networks — transformers.

**(a) Generalization gap:** In Figure 5a, we plot the generalization gap, which is test loss minus training loss on each labeled dataset, of self-distillation and further pre-training method. Self-distillation improves the generalization gap of the further pre-training method across all the datasets. It is consistent with Theorem 1 showing that self-distillation with a simplified model strictly decreases the generalization bound on the supervised loss of the fine-tuning stage.

**(b) Effect of self-distillation on distance:** To empirically validate Theorem 2 about regularization effects by self-distillation on $L_2$ distance between the initial pre-trained weight $\theta_{\texttt{init}}$ and the final weight after fine-tuning, we plot the distance obtained from self-distillation and further pre-training. Specifically, we compare the distance $\|\theta_{\texttt{init}} - \theta_{1,T}\|_2$ and $\|\theta_{\texttt{init}} - \theta_{0,T}\|_2$, where $\theta_{t,\tau}$ is the parameter after $t$ round of self-distillation and $\tau$ steps of gradient descent for fine-tuning. As shown in Figure 5b, self-distillation consistently decreases the distance and the reduced distance correlates with the better generalization gap in Figure 5a. These empirical results confirm the connection between the $L_2$ distance from the initialization and generalization bound (Nagarajan & Kolter, 2019).

**(c) Effect of multi-round self-distillation:** Lastly, we empirically verify part of Theorem 2 which shows that the first round of self-distillation plays the most critical role of regularization on the $L_2$ distance between the initial pre-trained weight $\theta_{\texttt{init}}$ and the final weight $\theta_{t,T}$ denoted as the parameter after $t$ round of self-distillation and $T$ steps of gradient descent for fine-tuning on VGG flower 102 dataset. As shown in Figure 5c, self-distillation significantly decreases the distance at the first round ($t = 1$) and the regularization effect on the distance diminishes afterward, where 0 round of self-distillation ($t = 0$) denotes the model with further pre-training but without self-distillation.

## 6 CONCLUSION

To effectively adapt pre-trained transformers to a target domain, we proposed self-distillation as a regularization for further pre-training. Specifically, we first took the initial pre-trained transformer and continued to pre-train it with the masked auto-encoding objective on the target unlabeled dataset and considered the encoder part of the model as a teacher for self-distillation. Then we took the copy of the same initial pre-trained model as a student and enforced representations of the student to be close to those of the teacher while optimizing the student with the masked auto-encoding objective on the target unlabeled dataset. Finally, we fine-tuned the self-distilled student on the target labeled dataset. Our empirical evaluation on various image and text classification benchmark datasets showed that self-distillation consistently improved generalization performance compared to relevant baselines. Lastly, we provided the theoretical analysis of the proposed method with a simplified model to understand how self-distillation for further pre-training can potentially help improve the generalization performance of the downstream tasks.

## REPRODUCIBILITY STATEMENT

We use Pytorch (Paszke et al., 2019) and transformers library (Wolf et al., 2020) from Huggingface to implement all the baselines and our proposed method in the experiments. We have described our method of self-distillation for further pre-training in Algorithm 1 and specified all the experimental setup including hyperparameters in Section 5 and Appendix E. For theoretical analysis, we have provided all the proofs in Appendix A.

## ACKNOWLEDGMENTS

This work was supported by Institute of Information & communications Technology Planning & Evaluation (IITP) grant funded by the Korea government(MSIT) (No.2019-0-00075, Artificial Intelligence Graduate School Program(KAIST)), the Engineering Research Center Program through the National Research Foundation of Korea (NRF) funded by the Korean Government MSIT (NRF-2018R1A5A1059921), Institute of Information & communications Technology Planning & Evaluation (IITP) grant funded by the Korea government(MSIT) (No. 2021-0-02068, Artificial Intelligence Innovation Hub), Institute of Information & communications Technology Planning & Evaluation (IITP) grant funded by the Korea government(MSIT) (No. 2022-0-00184, Development and Study of AI Technologies to Inexpensively Conform to Evolving Policy on Ethics), Institute of Information & communications Technology Planning & Evaluation (IITP) grant funded by the Korea government(MSIT) (No.2022-0-00713), KAIST-NAVER Hypercreative AI Center, and Samsung Electronics (IO201214-08145-01).

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

## Appendix

## A   Proofs

We also define the model output vector $f_t \in \mathbb{R}^{n \times p}$ by $(f_t)_{ij} = f(x_i, w_t)_j$. For example, $f_0$ is the initial teacher label matrix. Let $[n] = \{1, \ldots, n\}$. Denote the rank of $[I_p \otimes \Phi]$ by $r = \mathrm{rank}([I_p \otimes \Phi]) \le np$. Define $\tilde{U} = [u_1, u_2 \ldots, u_r] \in \mathbb{R}^{dp \times r}$ and $\mathbf{P}_r = I - \tilde{U}\tilde{U}^\top$, which is the projection matrix onto the null space of $\tilde{U}^\top$. We first prove the following lemma, which will be used in the proofs of Theorem 1 and Theorem 2 later:

**Lemma 1.** *For any* $t \in \mathbb{N}_0$, $w_{t,0} = \sum_{i=1}^r \alpha_{i,t}\tilde{u}_i + \mathbb{1}\{t = 0\}v$, *where* $\alpha_{i,t} = \frac{1}{\sigma_i}\left(\frac{1}{1+(n\lambda/\sigma_i^2)}\right)^t \in \mathbb{R}$, $\tilde{u}_i = \tilde{y}_i u_i \in \mathbb{R}^{dp}$, $\tilde{y}_i = (V^\top \mathrm{vec}[f_0])_i \in \mathbb{R}$, *and* $v = \mathbf{P}_r w_{0,0}$.

*Proof of Lemma 1.* Define $w_t := w_{t,0}$ for $t \in \mathbb{N}_+$. The necessary condition on the solution $w_t$ at step $t$ is that $\nabla L(w_t) = 0$. Thus, by solving $\nabla L(w_t) = 0$ for $w_t$, we have that $w_t = ([I_p \otimes \Phi][I_p \otimes \Phi]^\top + n\lambda I)^{-1}[I_p \otimes \Phi]\,\mathrm{vec}[f_{t-1}]$. By using the singular value decomposition $[I_p \otimes \Phi] = U\Sigma V^\top$, since $UU^\top = U^\top U = I$ and $V^\top V = I$, we have that

$$w_t = (U\Sigma\Sigma^\top U^\top + n\lambda I)^{-1}U\Sigma V^\top \mathrm{vec}[f_{t-1}] = (U(\Sigma\Sigma^\top + n\lambda I)U^\top)^{-1}U\Sigma V^\top \mathrm{vec}[f_{t-1}]$$
$$= U(\Sigma\Sigma^\top + n\lambda I)^{-1}\Sigma V^\top \mathrm{vec}[f_{t-1}].$$

Therefore, $w_t = U(\Sigma\Sigma^\top + n\lambda I)^{-1}\Sigma V^\top \mathrm{vec}[f_{t-1}]$. Using this and $[I_p \otimes \Phi] = U\Sigma V^\top$,

$$\mathrm{vec}[f_t] = \mathrm{vec}[\Phi^\top W_t^\top I_p] = [I_p \otimes \Phi]^\top w_t = [I_p \otimes \Phi]^\top U(\Sigma\Sigma^\top + n\lambda I)^{-1}\Sigma V^\top \mathrm{vec}[f_{t-1}]$$
$$= V\Sigma^\top(\Sigma\Sigma^\top + n\lambda I)^{-1}\Sigma V^\top \mathrm{vec}[f_{t-1}].$$

Therefore, $\mathrm{vec}[f_t] = VAV^\top \mathrm{vec}[f_{t-1}]$ where $A = \Sigma^\top(\Sigma\Sigma^\top + n\lambda I)^{-1}\Sigma$. Repeating this process for $\mathrm{vec}[f_{t-1}]$, since $V^\top V = I$,

$$\mathrm{vec}[f_t] = VAV^\top VAV^\top \cdots VAV^\top \mathrm{vec}[f_0] = VA^t V^\top \mathrm{vec}[f_0].$$

Plugging this equation of $\mathrm{vec}[f_{t-1}] = VA^{t-1}V^\top \mathrm{vec}[f_0]$ into the equation of $w_t = U(\Sigma\Sigma^\top + n\lambda I)^{-1}\Sigma V^\top \mathrm{vec}[f_{t-1}]$, we have that

$$w_t = U(\Sigma\Sigma^\top + n\lambda I)^{-1}\Sigma V^\top V A^{t-1}V^\top \mathrm{vec}[f_0] = UBA^{t-1}V^\top \mathrm{vec}[f_0]$$

where $B = (\Sigma\Sigma^\top + n\lambda I)^{-1}\Sigma$. Here, we can rewrite the matrix $B \in \mathbb{R}^{dp \times np}$ as

$$B = \begin{bmatrix} \bar{B} \\ \mathbf{0}_{(dp-np) \times np} \end{bmatrix}$$

where $\mathbf{0}_{(dp-np) \times np}$ is the $(dp - np)$ by $np$ matrix with all entries being zero, and $\bar{B} \in \mathbb{R}^{np \times np}$ is a diagonal matrix defined by

$$\bar{B}_{ii} := \sigma_i(\sigma_i^2 + n\lambda)^{-1}.$$

Using this $B$ in the above equation of $w_t = UBA^{t-1}V^\top \mathrm{vec}[f_0]$,

$$w_t = U\begin{bmatrix} \bar{B} \\ \mathbf{0}_{(dp-np) \times np} \end{bmatrix} A^{t-1}V^\top \mathrm{vec}[f_0]$$
$$= [u_1 \quad u_2 \quad \cdots \quad u_{dp}]\begin{bmatrix} \bar{B}A^{t-1} \\ \mathbf{0}_{(dp-np) \times np} \end{bmatrix} V^\top \mathrm{vec}[f_0]$$
$$= \bar{U}\bar{B}A^{t-1}V^\top \mathrm{vec}[f_0]$$

where $\bar{U} = [u_1, u_2 \ldots, u_{np}] \in \mathbb{R}^{dp \times np}$. Since the matrix $A = \Sigma^\top(\Sigma\Sigma^\top + n\lambda I)^{-1}\Sigma \in \mathbb{R}^{np \times np}$ is a diagonal matrix with its entry being $A_{ii} = \sigma_i^2(\sigma_i^2 + n\lambda)^{-1}$, this can be further simplified as

$$w_t = \bar{U}\bar{B}A^{t-1}V^\top \text{vec}[f_0]$$

$$= \bar{U}\begin{bmatrix} \frac{\sigma_1^1}{\sigma_1^2+n\lambda} & & \\ & \ddots & \\ & & \frac{\sigma_{np}^1}{\sigma_{np}^2+n\lambda} \end{bmatrix}\begin{bmatrix} \frac{\sigma_1^2}{\sigma_1^2+n\lambda} & & \\ & \ddots & \\ & & \frac{\sigma_{np}^2}{\sigma_{np}^2+n\lambda} \end{bmatrix}^{t-1}\begin{bmatrix} \tilde{y}_1 \\ \tilde{y}_2 \\ \vdots \\ \tilde{y}_{np} \end{bmatrix}$$

$$= \begin{bmatrix} u_1 & u_2 & \cdots & u_{np} \end{bmatrix}\begin{bmatrix} \sigma_1(\sigma_1^2+n\lambda)^{-1}(\sigma_1^2(\sigma_1^2+n\lambda)^{-1})^{t-1}\tilde{y}_1 \\ \sigma_2(\sigma_2^2+n\lambda)^{-1}(\sigma_2^2(\sigma_2^2+n\lambda)^{-1})^{t-1}\tilde{y}_2 \\ \vdots \\ \sigma_{np}(\sigma_{np}^2+n\lambda)^{-1}(\sigma_{np}^2(\sigma_{np}^2+n\lambda)^{-1})^{t-1}\tilde{y}_{np} \end{bmatrix}$$

$$= \sum_{i=1}^{np} \sigma_i(\sigma_i^2+n\lambda)^{-1}(\sigma_i^2(\sigma_i^2+n\lambda)^{-1})^{t-1}\tilde{y}_i u_i$$

$$= \sum_{i=1}^{r} \sigma_i(\sigma_i^2+n\lambda)^{-1}(\sigma_i^2(\sigma_i^2+n\lambda)^{-1})^{t-1}\tilde{y}_i u_i$$

where the last line follows from the fact that $\sigma_i(\sigma_i^2+n\lambda)^{-1}(\sigma_i^2(\sigma_i^2+n\lambda)^{-1})^{t-1} = 0$ for all $i > r$. Since $\sigma_i(\sigma_i^2+n\lambda)^{-1}(\sigma_i^2(\sigma_i^2+n\lambda)^{-1})^{t-1} = \sigma_i^{2t-1}(\sigma_i^2+n\lambda)^{-t} = \frac{1}{\sigma_i}(\frac{\sigma_i^2}{\sigma_i^2+n\lambda})^t$ for $i \le r$, this implies that

$$w_t = \sum_{i=1}^{r} \frac{1}{\sigma_i}\left(\frac{\sigma_i^2}{\sigma_i^2+n\lambda}\right)^t \tilde{y}_i u_i = \sum_{i=1}^{r} \frac{1}{\sigma_i}\left(\frac{1}{1+(n\lambda/\sigma_i^2)}\right)^t \tilde{y}_i u_i$$

Since $t \in \mathbb{N}_+$ was arbitrary, this holds for any $t \in \mathbb{N}_+$. This proves the first statement of the theorem for any $t \in \mathbb{N}_+$. For $t = 0$, since

$$\tilde{y}_i = (V^\top \text{vec}[f_0])_i = (V^\top[I_p \otimes \Phi]^\top w_{0,0})_i = (V^\top V\Sigma^\top U^\top w_{0,0})_i = (\Sigma^\top U^\top w_{0,0})_i = \sigma_i u_i^\top w_{0,0},$$

we have that

$$\sum_{i=1}^{r} \frac{1}{\sigma_i}\left(\frac{1}{1+(n\lambda/\sigma_i^2)}\right)^t \tilde{y}_i u_i = \left(\sum_{i=1}^{r} u_i u_i^\top\right) w_{0,0} = \tilde{U}\tilde{U}^\top w_{0,0}.$$

Thus,

$$w_{0,0} = \tilde{U}\tilde{U}^\top w_{0,0} + (I - \tilde{U}\tilde{U}^\top)w_{0,0} = \sum_{i=1}^{r} \frac{1}{\sigma_i}\left(\frac{1}{1+(n\lambda/\sigma_i^2)}\right)^t \tilde{y}_i u_i + (I - \tilde{U}\tilde{U}^\top)w_{0,0}.$$

Since $(I - \tilde{U}\tilde{U}^\top)w_{0,0} = \mathbf{P}_r w_{0,0}$, this completes the first statement of the theorem for any $t \in \mathbb{N}_0$.

$\square$

### A.1 PROOF OF THEOREM 1

*Proof.* Define $Z := [I_p \otimes \Phi]^\top \in \mathbb{R}^{\bar{n} \times \bar{d}}$ where $\bar{n} = np$ and $\bar{d} = dp$. Then,

$$\mathcal{L}(w) = \frac{1}{2}\sum_{i=1}^{n} \|f(x_i, w) - y_i\|_2^2 = \frac{1}{2}\|Zw - Y\|_2^2$$

where $Y = \text{vec}[[y_1, \ldots, y_n]^\top] \in \mathbb{R}^{\bar{n}}$. Since $\nabla\mathcal{L}(w_{t,\tau}) = Z^\top(Zw_{t,\tau} - Y)$,

$$\frac{dw_{t,\tau}}{d\tau} = -Z^\top(Zw_{t,\tau} - Y)$$

Since $\text{rank}(\Phi) = n$ and $d \ge n$, we have $\text{rank}(Z) = \bar{n}$ by the property of the Kronecker product with the identity matrix. Since $\text{rank}(Z) = \bar{n}$, there exists $v \in \mathbb{R}^{\bar{d}}$ such that $Y = Zv$. Thus,

$$\frac{dw_{t,\tau}}{d\tau} = -Z^\top(Zw_{t,\tau} - Zv)$$

$$= -Z^\top Z(w_{t,\tau} - v)$$

$$= -Z^\top Z(w_{t,\tau} - v).$$

Since $Z^\top = U\Sigma V^\top$, we have $Z^\top Z = U\Sigma\Sigma^\top U^\top = \sum_{i=1}^{\bar{n}} \sigma_i^2 u_i u_i^\top$. Thus,

$$\frac{dw_{t,\tau}}{d\tau} = -\left(\sum_{i=1}^{\bar{n}} \sigma_i^2 u_i u_i^\top\right)(w_{t,\tau} - v) = -\sum_{i=1}^{\bar{n}} \sigma_i^2 u_i u_i^\top (w_{t,\tau} - v).$$

Since the columns of $U$ forms the basis of $\mathbb{R}^{\bar{d}}$ and $w, v \in \mathbb{R}^{\bar{d}}$, we can write $w_{t,\tau} = \sum_{k=1}^{\bar{d}} c_k^{(t,\tau)} u_k$ and $v = \sum_{k=1}^{\bar{d}} q_k u_k$ for some $c_k^{(t,\tau)}$ and $q_k$. Thus,

$$\frac{dw_{t,\tau}}{d\tau} = -\sum_{i=1}^{\bar{n}} \sigma_i^2 u_i u_i^\top \sum_{k=1}^{\bar{d}} (c_k^{(t,\tau)} - q_k) u_k$$

$$= -\sum_{i=1}^{\bar{n}} \sum_{k=1}^{\bar{d}} \sigma_i^2 (c_k^{(t,\tau)} - q_k) u_i u_i^\top u_k$$

$$= -\sum_{i=1}^{\bar{n}} \sigma_i^2 (c_i^{(t,\tau)} - q_i) u_i.$$

Using $w_{t,\tau} = \sum_{k=1}^{\bar{d}} c_k^{(t,\tau)} u_k$ for the right-hand side too, we have that

$$\frac{d}{d\tau} \sum_{i=1}^{\bar{d}} c_i^{(t,\tau)} u_i = -\sum_{i=1}^{\bar{n}} \sigma_i^2 (c_i^{(t,\tau)} - q_i) u_i.$$

This implies that for all $i \in \{1, \ldots, \bar{n}\}$,

$$\frac{d}{d\tau} c_i^{(t,\tau)} = -\sigma_i^2 (c_i^{(t,\tau)} - q_i),$$

and $\frac{d}{d\tau} c_i^{(t,\tau)} = 0$ for all $i \notin \{1, \ldots, \bar{n}\}$. This can be also seen by the fact that $\frac{dw_{t,\tau}}{d\tau} = -Z^\top(Zw_{t,\tau} - Zv)$ with $Z^\top = U\Sigma V^\top$ and thus the dynamics only adds components of $u_i$ for $i \in \{1, \ldots, \bar{n}\}$, and not for $i \notin \{1, \ldots, \bar{n}\}$. Thus, for components of $u_i$ for $i \notin \{1, \ldots, \bar{n}\}$, the initial values stays. In other words, for $i \notin \{1, \ldots, \bar{n}\}$,

$$c_i^{(t,\tau)} = c_i^{(t,0)}.$$

On the other hand, for $i \in \{1, \ldots, \bar{n}\}$, since $\frac{d}{d\tau} q_i = 0$,

$$\frac{d}{d\tau}(c_i^{(t,\tau)} - q_i) = \frac{d}{d\tau} c_i^{(t,\tau)} = -\sigma_i^2 (c_i^{(t,\tau)} - q_i).$$

Solving this for $(c_i^{(t,\tau)} - q_i)$, we have that for $i \in \{1, \ldots, \bar{n}\}$,

$$c_i^{(t,\tau)} - q_i = (c_i^{(t,0)} - q_i) e^{-\sigma_i^2 \tau}.$$

This implies that

$$c_i^{(t,\tau)} = q_i + (c_i^{(t,0)} - q_i) e^{-\sigma_i^2 \tau} = q_i(1 - e^{-\sigma_i^2 \tau}) + c_i^{(t,0)} e^{-\sigma_i^2 \tau}.$$

Combining these with $w_{t,T} = \sum_{k=1}^{\bar{d}} c_k^{(t,T)} u_k$,

$$w_{t,T} = \sum_{i=1}^{\bar{d}} c_i^{(t,T)} u_i = \sum_{i=1}^{\bar{n}} q_i(1 - e^{-\sigma_i^2 T}) u_i + \sum_{i=1}^{\bar{n}} c_i^{(t,0)} e^{-\sigma_i^2 T} u_i + \sum_{i=\bar{n}+1}^{\bar{d}} c_i^{(t,0)} u_i. \quad (5)$$

Therefore, for any particular $s \in \mathcal{S}$, since $U = [u_1, u_2 \ldots, u_{dp}] \in \mathbb{R}^{dp \times dp}$ is an orthogonal matrix,

$$\|\mathcal{A}_t(s)\|_2^2 = \|w_{t,T}\|_2^2 \leq \sum_{i=1}^{\bar{n}} \left(q_i(1 - e^{-\sigma_i^2 T})\right)^2 + \sum_{i=1}^{\bar{n}} (c_i^{(t,0)})^2 e^{-2\sigma_i^2 T} + \sum_{i=\bar{n}+1}^{\bar{d}} (c_i^{(t,0)})^2. \quad (6)$$

where $q_i, \sigma_i$, and $c_i^{(t,0)}$ all depend on $s$.

By using Lemma 4 of (Pham et al., 2021) and taking union bound with $\mathbb{P}(s \notin \mathcal{S}) \leq \delta$, with probability at least $1 - \delta$, we have that $w_{t,T} \in \mathcal{F}_t$ and the following holds:

$$\mathbb{E}_{x,y}[\ell(w_{t,T}, x, y)] \leq \frac{1}{n} \sum_{i=1}^{n} \ell(w_{t,T}, x_i, y_i) + 2\mathcal{R}_n(\mathcal{F}_t) + M\sqrt{\frac{\ln(2/\delta)}{2n}}, \tag{7}$$

where $\mathcal{R}_n(\mathcal{F}_t) = \mathbb{E}_{s,\xi}[\sup_{w \in \mathcal{F}_t} \frac{1}{n} \sum_{i=1}^{n} \xi_i \|W\varphi(x_i) - y_i\|_2^2]$, $s = ((x_i, y_i))_{i=1}^{n}$, $w = \text{vec}[W^\top]$, and $\xi_1, \ldots, \xi_n$ are independent uniform random variables taking values in $\{-1, 1\}$. By using Corollary 4 of (Maurer, 2016), there exits a constant $c$ (only depending on $M$) such that,

$$\mathcal{R}_n(\mathcal{F}_t) \leq \frac{c}{n} \mathbb{E}_{s,\xi}[\sup_{w \in \mathcal{F}_t} \sum_{i=1}^{n} \sum_{k=1}^{p} \xi_{ik} W_k \varphi(x_i)]$$

$$= \frac{c}{n} \mathbb{E}_{s,\xi}[\sup_{w \in \mathcal{F}_t} \sum_{k=1}^{p} W_k \sum_{i=1}^{n} \xi_{ik} \varphi(x_i)]$$

$$= \frac{c}{n} \mathbb{E}_{s,\xi}[\sup_{w \in \mathcal{F}_t} w^\top h]$$

where $W_k$ is the $k$-th row of $W$, $\xi_{ik}$ are independent uniform random variables taking values in $\{-1, 1\}$, $h = \text{vec}[H] \in \mathbb{R}^{dp}$, and $H \in \mathbb{R}^{d \times p}$ with $H_{jk} = \sum_{i=1}^{n} \xi_{ik} \varphi(x_i)_j$. Thus,

$$\mathcal{R}_n(\mathcal{F}_t) \leq \frac{c}{n} \mathbb{E}_{s,\xi}[\sup_{w \in \mathcal{F}_t} \|w\|_2 \|h\|_2] = \frac{c(\sup_{w \in \mathcal{F}_t} \|w\|_2)}{n} \mathbb{E}_{s,\xi}[\|h\|_2]$$

Here,

$$\mathbb{E}_{s,\xi}[\|h\|_2] = \mathbb{E}_{s,\xi} \sqrt{\sum_{j=1}^{d} \sum_{k=1}^{p} \left(\sum_{i=1}^{n} \xi_{ik} \varphi(x_i)_j\right)^2} \leq \sqrt{\sum_{j=1}^{d} \sum_{k=1}^{p} \mathbb{E}_{s,\xi} \left(\sum_{i=1}^{n} \xi_{ik} \varphi(x_i)_j\right)^2}$$

$$= \sqrt{\sum_{j=1}^{d} \sum_{k=1}^{p} \mathbb{E}_s \sum_{i=1}^{n} (\varphi(x_i)_j)^2} \tag{8}$$

$$= \sqrt{\sum_{k=1}^{p} \sum_{i=1}^{n} \mathbb{E}_s \sum_{j=1}^{d} (\varphi(x_i)_j)^2}$$

$$= \sqrt{\sum_{k=1}^{p} \sum_{i=1}^{n} \mathbb{E}_s \|\varphi(x_i)\|_2^2}$$

$$\leq R\sqrt{pn}$$

Equation 8 holds since

$$\mathbb{E}_{s,\xi}[(\xi_{ik}\phi(x_i)_j) \cdot (\xi_{lk}\phi(x_l)_j)] = \mathbb{E}_s[\mathbb{1}\{i = l\}\phi(x_i)_j\phi(x_l)_j]$$

for all $i, l \in [n]$.

Thus,

$$\mathcal{R}_n(\mathcal{F}_t) \leq \frac{cR\sqrt{p}(\sup_{w \in \mathcal{F}_t} \|w\|_2)}{\sqrt{n}}. \tag{9}$$

Define

$$\zeta_t(s) := \sqrt{\sum_{i=1}^{\bar{n}} \left(q_i(1 - e^{-\sigma_i^2 T})\right)^2 + \sum_{i=1}^{\bar{n}} (c_i^{(t,0)})^2 e^{-2\sigma_i^2 T} + \sum_{i=\bar{n}+1}^{\bar{d}} (c_i^{(t,0)})^2}.$$

where $q_i, \sigma_i$, and $c_i^{(t,0)}$ all depend on $s$. With this, we define

$$\zeta(t) := \sup_{s \in \mathcal{S}} \zeta_t(s).$$

Then, by combining equation 6, equation 7, and equation 9, with probability at least $1 - \delta$, the following holds:

$$\mathbb{E}_{x,y}[\ell(w_{t,T}, x, y)] \leq \frac{1}{n}\sum_{i=1}^{n}\ell(w_{t,T}, x_i, y_i) + \zeta(t)\sqrt{\frac{4c^2R^2p}{n}} + M\sqrt{\frac{\ln(2/\delta)}{2n}}.$$

Finally, from Lemma 1, for any $t \in \mathbb{N}_0$ and $i \in \{1, \ldots, \bar{n}\}$,

$$(c_i^{(t,0)})^2 = \left(\frac{1}{\sigma_i}\left(\frac{1}{1 + (n\lambda/\sigma_i^2)}\right)^t \tilde{y}_i\right)^2.$$

Since $\frac{1}{1+(n\lambda/\sigma_i^2)} < 1$ (because $n\lambda/\sigma_i^2 > 0$), the value of $\left(\frac{1}{1+(n\lambda/\sigma_i^2)}\right)^{2t}$ strictly decreases as $t$ increases. Since $\frac{1}{\sigma_i^2} > 0$ and $\tilde{y}_i^2 \geq 0$, this implies that $(c_i^{(t,0)})^2$ is strictly decreasing in $t \in \mathbb{N}_0$ unless $c_i^{(t,0)} = 0$. Moreover, from Lemma 1, we have that

$$w_{t,0} = \sum_{i=1}^{\bar{n}}\alpha_{i,t}\tilde{y}_i u_i + \mathbb{1}\{t = 0\}(I - \tilde{U}\tilde{U}^\top)w_{0,0}.$$

Since $\{u_1, \ldots, u_{\bar{d}}\}$ is a orthonormal basis for $\mathbb{R}^{\bar{d}}$ with inner product $\langle x, y \rangle = y^\top x$, we get

$$w_{0,0} = \sum_{i=1}^{\bar{d}}(u_i^\top w_{0,0})u_i.$$

Since $\tilde{U}\tilde{U}^\top w_{0,0} = \sum_{i=1}^{\bar{n}}(u_i^\top w_{0,0})u_i$, we have that

$$(I - \tilde{U}\tilde{U}^\top)w_{0,0} = \sum_{i=1}^{\bar{d}}(u_i^\top w_{0,0})u_i - \sum_{i=1}^{\bar{n}}(u_i^\top w_{0,0})u_i = \sum_{i=\bar{n}+1}^{\bar{d}}(u_i^\top w_{0,0})u_i,$$

which implies that the $u_i$ component to span $w_{t,0}$ for $i \in \{\bar{n} + 1, \ldots, \bar{d}\}$ is only present in $(I - \tilde{U}\tilde{U}^\top)w_{0,0}$. In other words,

$$w_{t,0} = \sum_{i=1}^{\bar{n}}\alpha_{i,t}\tilde{y}_i u_i + \sum_{i=\bar{n}+1}^{\bar{d}}\mathbb{1}\{t = 0\}(u_i^\top w_{0,0})u_i.$$

Thus, for any $t \in \mathbb{N}_0$ and $i \in \{\bar{n} + 1, \ldots, \bar{d}\}$, we have that

$$(c_i^{(t,0)})^2 = \mathbb{1}\{t = 0\}(u_i^\top w_{0,0})^2.$$

These implies that $\zeta(t)$ is strictly decreasing in $t \in \mathbb{N}_0$ unless $w_{0,0} = 0$. $\qquad\square$

## A.2 PROOF OF THEOREM 2

*Proof.* In this proof, we continue to use the results and the notation from the proof of Theorem 1. By using equation 5 in the proof of Theorem 1, we have that

$$\|w_{\text{init}} - w_{t,T}\|_2 = \|w_{\text{init}} - v_t\|_2,$$

where

$$v_t = \sum_{i=1}^{\bar{n}}q_i(1 - e^{-\sigma_i^2 T})u_i + \sum_{i=1}^{\bar{n}}c_i^{(t,0)}e^{-\sigma_i^2 T}u_i + \sum_{i=\bar{n}+1}^{\bar{d}}c_i^{(t,0)}u_i.$$

If $w_{\text{init}} = -\alpha v_t$ for some $\alpha > 0$, then

$$\begin{aligned}
\|w_{\text{init}} - v_t\|_2 &= \|v_t + \alpha v_t\|_2 \\
&= (1 + \alpha)\|v_t\|_2 \\
&= \|v_t\|_2 + \|\alpha v_t\|_2 \\
&= \sqrt{\sum_{i=1}^{\bar{n}}q_i^2(1 - e^{-\sigma_i^2 T})^2 + \sum_{i=1}^{\bar{n}}(c_i^{(t,0)})^2 e^{-2\sigma_i^2 T} + \sum_{i=\bar{n}+1}^{\bar{d}}(c_i^{(t,0)})^2} + \|w_{\text{init}}\|_2.
\end{aligned}$$

On the other hand, for any $w_{\text{init}} \in \mathbb{R}^{dp}$,

$$\|w_{\text{init}} - v_t\|_2 \leq \|v_t\|_2 + \|w_{\text{init}}\|_2$$

$$\leq \sqrt{\sum_{i=1}^{\bar{n}} q_i^2 (1 - e^{-\sigma_i^2 T})^2 + \sum_{i=1}^{\bar{n}} (c_i^{(t,0)})^2 e^{-2\sigma_i^2 T} + \sum_{i=\bar{n}+1}^{\bar{d}} (c_i^{(t,0)})^2} + \|w_{\text{init}}\|_2.$$

Thus, setting $\psi(t)$ to be the following function satisfies conditions (1) and (2) in the statement:

$$\psi(t) := \sqrt{\sum_{i=1}^{\bar{n}} q_i^2 (1 - e^{-\sigma_i^2 T})^2 + \sum_{i=1}^{\bar{n}} (c_i^{(t,0)})^2 e^{-2\sigma_i^2 T} + \sum_{i=\bar{n}+1}^{\bar{d}} (c_i^{(t,0)})^2} + \|w_{\text{init}}\|_2$$

Finally, from Lemma 1, for any $t \in \mathbb{N}_0$ and $i \in \{1, \ldots, \bar{n}\}$,

$$(c_i^{(t,0)})^2 = \left( \frac{1}{\sigma_i} \left( \frac{1}{1 + (n\lambda/\sigma_i^2)} \right)^t \tilde{y}_i \right)^2.$$

which is strictly decreasing in $t \in \mathbb{N}_0$ unless $c_i^{(t,0)} = 0$ for all $i \in \{1, \ldots, \bar{n}\}$ as shown in the proof of Theorem 1. Moreover, from Lemma 1, for any $t \in \mathbb{N}_0$ and $i \in \{\bar{n}+1, \ldots, \bar{d}\}$,

$$(c_i^{(t,0)})^2 = \mathbb{1}\{t = 0\}(u_i^\top w_{0,0})^2.$$

That is,

$$\psi(t) = \sqrt{G_1 + \psi_1(t) + \sum_{i=\bar{n}+1}^{\bar{d}} \mathbb{1}\{t = 0\}(u_i^\top w_{0,0})^2} + G_2,$$

where

$$G_1 := \sum_{i=1}^{\bar{n}} q_i^2 (1 - e^{-\sigma_i^2 T})^2,$$

$$\psi_1(t) := \sum_{i=1}^{\bar{n}} \left( \frac{1}{\sigma_i} \left( \frac{1}{1 + (n\lambda/\sigma_i^2)} \right)^t \tilde{y}_i \right)^2 e^{-2\sigma_i^2 T},$$

and

$$G_2 := \|w_{\text{init}}\|_2.$$

Since $e^{-2\sigma_i^2 T} > 0$ is a constant in $t$ and we have previously shown that $\left( \frac{1}{\sigma_i} \left( \frac{1}{1+(n\lambda/\sigma_i^2)} \right)^t \tilde{y}_i \right)^2$ is strictly decreasing in $t \in \mathbb{N}_0$ unless $w_{0,0} = 0$. It implies that both $\psi_1(t)$ and $\psi(t)$ are strictly decreasing in $t \in \mathbb{N}_0$ unless $w_{0,0} = 0$. □

**Remark 1.** Note that Theorem 2 also shows the distance between the weight of the teacher $w_{t-1,T}$ and the initial pre-trained weight $w_{\text{init}}$ for all $t \in \mathbb{N}_+$. Since the teacher at $t' \in \mathbb{N}_+$ round of self-distillation used to be a student of the $t' - 1$ round of the self-distillation and Theorem 2 holds for all non-negative integer $t \in \mathbb{N}_0$, the distance between the initial weight and the teacher weight $\|w_{\text{init}} - w_{t-1,T}\|_2$ strictly decreases for all $t \in \mathbb{N}_+$. For instance, at $t = 1$, we obtain the following inequality

$$\|w_{\text{init}} - w_{0,T}\|_2 \leq \psi(0),$$

where $w_{0,T}$ is the weight of the initial teacher *without self-distillation*.

## B  ADDITIONAL EXPERIMENTS

In this section, we perform additional experiments to better analyze the proposed method, self-distillation for further pre-training.

Table 4: Training and test loss.

| Method | Split | Aircraft | CUB | Chest-Xray | DTD | Dogs | Flower |
|---|---|---|---|---|---|---|---|
| Further Pre-training | Train (log) | $-9.13 \pm 0.09$ | $-8.58 \pm 0.05$ | $-8.10 \pm 1.90$ | $-3.70 \pm 0.37$ | $-5.41 \pm 0.13$ | $-9.70 \pm 0.32$ |
| | Test | $1.44 \pm 0.03$ | $2.44 \pm 0.02$ | $2.26 \pm 0.22$ | $2.49 \pm 0.08$ | $1.96 \pm 0.02$ | $0.63 \pm 0.04$ |
| Self-Distillation | Train (log) | $-9.17 \pm 0.08$ | $-8.16 \pm 0.31$ | $-6.08 \pm 0.50$ | $-4.08 \pm 0.30$ | $-5.50 \pm 0.06$ | $-7.98 \pm 1.28$ |
| | Test | $\mathbf{1.36 \pm 0.01}$ | $\mathbf{2.26 \pm 0.04}$ | $\mathbf{2.08 \pm 0.17}$ | $\mathbf{2.15 \pm 0.06}$ | $\mathbf{1.90 \pm 0.01}$ | $\mathbf{0.50 \pm 0.03}$ |

**Supervised Contrastive Learning**    For the main experiments on Section 5, we use the same objective function for pre-training and further pre-training. One may wonder what happens if we use different objective at further pre-training stage. In this experiment, we continue to pre-train the encoder of the pre-trained transformer (He et al., 2022) with supervised contrastive loss (Khosla et al., 2020) while fine-tuning a randomly initialized linear classifier with cross-entropy loss. As shown in Table 5, supervised contrastive learning (SupCon) significantly degrades the generalization performance of Vision Transformer on both X-ray and Flower datasets. Based on this experimental result, we conjecture that the transformer pre-trained with a masked auto-encoding objective might not be compatible with the contrastive loss and thus we may use the same objective for pre-training and further pre-training.

Table 5: Comparison with supervised contrastive learning.

| Model | X-ray | Flower |
|---|---|---|
| SupCon | $75.30 \pm 1.03$ | $76.70 \pm 1.76$ |
| Fine-tuning | $77.15 \pm 0.52$ | $88.78 \pm 0.65$ |
| Further Pre-training | $77.79 \pm 2.06$ | $88.63 \pm 0.35$ |
| Self-Distillation | $\mathbf{79.68 \pm 1.05}$ | $79.68 \pm 0.44$ |

**Generalization Gap**    As shown in Figure 5a, self-distillation decreases generalization gap compared to further pre-training. Additionally, we separately report training and test loss in Table 4. Both of further pre-training and self-distillation reach near zero training loss, but self-distillation achieves way lower test loss than further pre-training as a consequence of regularization induced by self-distillation.

**Down Weighting of Masked Auto-Encoding**    Although we fix both weight of self-distillation and masked auto-encoding objective to 1 in equation 3, we vary $\lambda \in (0, 1]$, the weight of masked auto-encoding objective $(\lambda \mathcal{L}_{\text{MAE}}(\theta, \phi; \mathcal{D}^u) + \mathcal{L}_{\text{Distill}}(\theta; \theta_0, \mathcal{D}^u))$ and report test accuracy of Vision Transformer with self-distillation on the CUB dataset. As shown in Table 6, our proposed method is insensitive to the value of $\lambda$ and thus there is no benefit to fine-tuning the weight of masked auto-encoding objective.

Table 6: Down weighting MAE objective.

| $\lambda$ | CUB |
|---|---|
| 1.0 | $58.06 \pm 0.90$ |
| 0.5 | $57.76 \pm 0.17$ |
| 0.3 | $58.21 \pm 0.42$ |
| 0.1 | $57.76 \pm 0.33$ |

**Extra Training Time**    To better analyze extra computational cost for further pre-training and self-distillation, we report training wall clock time for fine-tuning, further pre-training, and self-distillation, respectively. We train a Vision Transformer (Dosovitskiy et al., 2021) on CUB dataset with 3090 RTX GPU and Intel(R) Xeon(R) Silver 4210R CPU. It takes 32 minutes and 18 seconds for fine-tuning the transformer. We need an extra 1 hour 29 minutes 33 seconds for further pre-training and 5 hours 13 minutes 22 seconds for self-distillation.

## C   MASKED AUTO-ENCODING

In this section, we describe the masked auto-encoding objective from equation 2 in more detail. Given a sequence $\mathbf{x} = (x_1, \ldots, x_K)$ with length $K$, we sample mask $\mathbf{z} = (z_1, \ldots, z_K)$ from a Binomial distribution $p_{\gamma, K}$ with probability for success $\gamma \in (0, 1)$ and the number of trials $K$. For each $x_k$, we replace it with the special token "mask" if $z_k = 1$. Otherwise we use the same $x_k$ for an masked input. Let $\hat{\mathbf{x}} = (\hat{x}_1, \ldots, \hat{x}_K)$ be a masked input and let $f_\theta, g_\phi$ be encoder and decoder, respectively. We want to compute the log-likelihood of the reconstructed input $\sum_{k=1}^{K} z_k \log p_{\theta, \phi}(x_k | \hat{\mathbf{x}})$.

For language models, reconstruction of the masked input $\hat{x}_k$ is predicting which token is masked out of pre-defined vocabulary with its size $V$, where each token is represented as an integer from $\{1, \ldots, V\}$. Thus the conditional probability of $x_k \in \{1, \ldots, V\}$ given $\hat{\mathbf{x}}$ is parameterized as

follows:

$$p_{\theta,\phi}(x_k|\hat{\mathbf{x}}) = \frac{\exp(u_{x_k}))}{\sum_{j=1}^{V} \exp(u_j)}$$

$$\text{where } (u_1, \ldots, u_V) = g_\phi(\mathbf{h}_k) \in \mathbb{R}^V, \quad \begin{bmatrix} | & & | \\ \mathbf{h}_1 & \cdots & \mathbf{h}_K \\ | & & | \end{bmatrix} = f_\theta(\hat{\mathbf{x}}) \in \mathbb{R}^{h \times K}.$$

For ViT, the sequence $\mathbf{x}$ consists of image patches and the reconstruction of the masked input is predicting pixel values for each masked patches, which is a regression problem. Thus, we parameterize the conditional probability of a patch $x_k = (x_{k,1}, \ldots, x_{k,m}) \in \mathbb{R}^m$ given $\hat{\mathbf{x}}$ as follows:

$$p_{\theta,\phi}(x_k|\hat{\mathbf{x}}) = \prod_{i=1}^{m} \frac{1}{\sqrt{2\pi\sigma^2}} \exp\left(-\frac{(x_{k,i} - \mu_{k,i})^2}{2\sigma^2}\right)$$

$$\text{where } \boldsymbol{\mu}_k = (\mu_{k,1}, \ldots, \mu_{k,m}) \in \mathbb{R}^m, \quad \begin{bmatrix} | & & | \\ \boldsymbol{\mu}_1 & \cdots & \boldsymbol{\mu}_K \\ | & & | \end{bmatrix} = f_\theta(\hat{\mathbf{x}}) \in \mathbb{R}^{m \times K}.$$

Since $\sigma > 0$ and $\pi$ are constants with respect to $\theta$ and $\phi$,

$$\arg\min_{\theta,\phi} -\sum_{k=1}^{K} \log p_{\theta,\phi}(x_k|\hat{\mathbf{x}}) = \arg\min_{\theta,\phi} \sum_{k=1}^{K} \left( \frac{1}{2\sigma^2} \sum_{j=1}^{m} (x_{k,j} - \mu_{k,j})^2 - m \log\left(\frac{1}{\sqrt{2\pi\sigma^2}}\right) \right)$$

$$= \arg\min_{\theta,\phi} \sum_{k=1}^{K} \left( \sum_{j=1}^{m} (x_{k,j} - \mu_{k,j})^2 \right)$$

$$= \arg\min_{\theta,\phi} \sum_{k=1}^{K} \|x_k - \boldsymbol{\mu}_k\|_2^2.$$

## D  DATASET

We describe statistics of all the image and text classification datasets used for our experiments in Table 7 and 8.

Table 7: The number of training instances and classes for each image classification dataset.

|  | Aircraft | CUB | Chest X-ray | DTD | Dogs | Flower |
|---|---|---|---|---|---|---|
| # of instances | 6,667 | 5,594 | 5,216 | 4,230 | 12,000 | 2,040 |
| # of classes | 100 | 200 | 2 | 47 | 120 | 102 |

Table 8: The number of training instances and classes for each text classification dataset.

|  | SCIERC | ACL-ARC | Chemprot | Twitter-Emotion |
|---|---|---|---|---|
| # of instances | 3,219 | 1,688 | 4,169 | 4,230 |
| # of classes | 7 | 6 | 13 | 47 |

# E   Hyperparameters

In Table 9, we summarize all the hyperparameters for Vision Transformer and RoBERTA.

Table 9: Hyperparameters for Vision Transformer and RoBERTA

| Hyperparameters | Vision Transformer | RoBERTA |
|---|---|---|
| learning rate for pre-training | $1.5 \cdot 10^{-4}$ | $1 \cdot 10^{-4}$ |
| learning rate for fine-tuning | $1 \cdot 10^{-4}$ | $2 \cdot 10^{-5}$ |
| weight decay coefficient | $1 \cdot 10^{-2}$ | $1 \cdot 10^{-2}$ |
| batch-size for pre-training | 64 | 128 |
| batch-size for fine-tuning | 32 | 16 |
| learning rate scheduler | linear-decay | linear-decay |
| fine-tuning steps | 10,000 | 10 epochs |
| pre-training steps | 20,000 | 100 epochs |
| round of self-distillation | 1 | 1 |

