# OpenReview forum: "Self-Distillation for Further Pre-training of Transformers"
_ICLR.cc/2023/Conference — ICLR 2023 poster_

### Official Review · Reviewer_7okZ · 2022-10-23

**Confidence:** 4
**Clarity, Quality, Novelty And Reproducibility:** The paper is very clear and well-writ…
**Correctness:** 4
**Technical Novelty And Significance:** 3
**Empirical Novelty And Significance:** 3
**Recommendation:** 6

**Details Of Ethics Concerns:**

N/A.

**Strength And Weaknesses:**

Strength.
+ I found the paper is very well written with a clear structure and motivation.
+ The design of self-distillation method is very clean and general. I can imagine it can be useful beyond image classification tasks.
+ The paper also provides theoretical analysis showing self-distillation acts as a regulariser on the distance between the initial pre-trained weights and final fine-tuned weights.
+ Ablative analysis is very comprehensive showing different design strategies within the self-distillation method.

Limitation.
- Further pre-training induces overfitting? The paper has highlighted multiple times that further pre-training would hurt transfer learning performance and overfits on the target dataset. However, from the Table 1 and 2, the further pre-training strategy actually performs typically the second right after the proposed method, which is a bit contractor to what the paper has highlighted, and it is always better than direct fine-tuning? I am wondering is this because this further pre-training is well tuned or some other reasons? A bit more information on this baseline would be important.
- A fairer comparison to the baseline. From Figure 2, we can see that the proposed self-distillation method requires 3 times longer than fine-tuning and 1.5 times longer than the further pre-training method.  Comparing with further pre-training might be okay since the further pre-training on the teacher model did not account to be included in the final performance evaluation. However, we may argue that it could be possible that the improved performance was due to longer training time. I would suggest having one small additional experiment showing that the self-distillation method works was due to the design rather than a longer training time.


**Summary Of The Paper:**

This paper proposes to use self-distillation as a regularization method to improve transfer learning performance on image classification tasks. The paper first observes that further fine-tuning on a pre-trained model on unlabelled target dataset would result overfitting. And then it proposes the self-distillation method to resolve this issue. The general idea of self-distillation method is to first further pre-train a teacher model on target unlabelled data with a MAE objective, and then guide a student model from the original pre-trained model for the same representation learned from the teacher model and additionally trained with the same MAE objective. And finally to fine-tune the student model with the labelled target data. The paper provides theorical analysis on the generalization bound for the proposed method and showed an improved performance over a range of fine-grained image classification benchmarks.

**Summary Of The Review:**

The paper overall is very good with theoretical analysis and very comprehensive experiment support. There are some minor issues listed in the limitation section and I would consider raising my score if the authors could clarify these issues.

---

> ### Author Response · Authors · 2022-11-09
> **Response to Reviewer 7okZ**
>
> [**Q1**] The design of the self-distillation method is very clean and general. I can imagine it can be useful beyond image classification tasks.
>
> - Thank you. Please recall that we have already performed experiments on **text classification** with pre-trained language models RoBERTA.
>
> ---
>
> [**Q2**] Does further pre-training induce overfitting? From Table 1 and 2, further pre-training is the second best performing model, which is always better than direct fine-tuning.
> - We respectfully disagree with you. This is not a contradiction. First, in the introduction, we have clearly stated that continual pre-training of Transformers (language models such as RoBERTA) on text data has been shown to be effective but  **Vision Transformer** is vulnerable to overfitting when we continue to pre-train it. Thus, the experimental results that further pre-training of **RoBERTA** is always better than fine-tuning do not contradict our claim.
> - Second, further pre-training of Vision Transformers is **not always better** than direct fine-tuning. In Table 1, further pre-training significantly degrades the performance of Vision Transformers ($67.56 \pm 0.52 \to 65.55 \pm 1.12$) on the DTD dataset. Moreover, other than the Aircraft dataset,  improvement of further pre-training is statistically not significant considering the standard deviation of multiple runs.
>
> ---
>
> [**Q3**] Since self-distillation trains a model longer than further pre-training, performing further pre-training longer would be a fair comparison to show that self-distillation works due to the regularization rather than a longer training time.
> - Thank you for the suggestion. We have already done this experiment for ablation study in section 5.1. As shown in Table 3, we further pre-train a transformer twice longer, denoted as **Further Pre-train $\times 2$**, and compare it to self-distillation. However, the model with self-distillation still outperforms the model with longer further pre-training.
>
> |                     | CUB       | SCIERC         |
> |------------------|-----------------|----------------|
> | Further Pre-train | $55.72 \pm 0.46$ | $80.32 \pm 1.25$ |
> | Further Pre-train $\times 2$              | $53.41 \pm 0.75$  | $80.52 \pm 0.98$ |
> | **Self-Distillation**          | $58.06 \pm 0.90 $  | $ 81.79 \pm 0.75$|
> [Part of Table3]

---

> > ### Comment · Reviewer_7okZ · 2022-11-26
> > **Response to the rebuttal**
> >
> > Thanks to the authors for the explanations. All my concerns have been resolved. I would love to keep my weak acceptance as my final rating.

---

### Official Review · Reviewer_RiKW · 2022-10-29

**Confidence:** 3
**Correctness:** 3
**Technical Novelty And Significance:** 2
**Empirical Novelty And Significance:** 3
**Recommendation:** 8

**Clarity, Quality, Novelty And Reproducibility:**

## Clarity
The paper is well written and clear.  I appreciate the prose explaining the main takeaways from the theoretical analysis.
## Quality
The empirical studies is conducted in a very thorough manner with multiple seeds, variance quantification, and extensive ablations.
## Reproducibility
I believe sufficient details are provided for the reader to be able to reproduce these experiments.

**Strength And Weaknesses:**

Pros:
- The empirical results for self-distillation relative to other forms of regularized fine-tuning is state-of-the art on both text and image classification tasks.
- The authors conduct a series of empirical studies and ablation studies to verify the regularization benefits of self-distillation.
- The writing is clear and well organized.

Cons:
- The approach is limited in novelty but I still think the proposed method is important.
- Figure 1 shows that even with self-distillation, the model regresses to no benefit from further pre-training after 60k steps.

Questions:
- From the text, it appears that self-distillation does not have a tunable parameter to trade off between MAE and Distill loss. Is there any benefit to tuning the weight for the two terms in the training objective?
- The theoretical analysis shows bounded distance between self-distilled weights and initial weights but does not comment on distance to teacher weights.  Is there a breakdown you can provide for that bound?


**Summary Of The Paper:**

This paper proposes a self-distillation approach to transferring a pre-trained model to a new task.  The method first applies further pre-training to arrive at a set of teacher weights that is then used to guide the subsequent student training starting from original pre-trained weights.  The authors show self-distillation with further pre-training to be an effective method for regularizing further pre-training for downstream tasks through a variety of empirical studies on image and text classification tasks.  Finally, theoretical analysis shows how self-distillation acts as a form of distance-based regularization on the initial pre-trained weights.

**Summary Of The Review:**

The authors introduce a simple yet effective way to transfer a pre-trained model to a new downstream task via self-distillation with further pre-training.  The experimental results are strong and show self-distillation to be state-of-the-art across multiple text and vision classification tasks.  Ablations also show self-distillation to be a critical component of these empirical results. Aside from a couple of questions and some concern regarding limited novelty, I would recommend the paper to be accepted due to its relevance and high potential for impact in practical workloads adapting a pre-trained model to a new task.

---

> ### Author Response · Authors · 2022-11-09
> **Response to Reviewer RiKW**
>
> [**Q1**] Figure 1 shows that even with self-distillation, the model regresses to no benefit from further pre-training after 60k steps.
>
> - Although self-distillation does not improve the generalization performance of ViT after 60k steps, it still **prevents ViT from overfitting** to the target dataset as a consequence of regularization induced by self-distillation.
>
> - Moreover,  the model trained with self-distillation for 60k steps significantly **outperforms the bad teacher** which is further pre-trained for 60k steps. Contrary to self-distillation, further pre-training with 60k steps significantly degrades the generalization performance of ViT.
>
> ---
>
> [**Q2**] Is there any benefit to tuning the weight for the two terms in the training objective?
> - Thank you for asking a good question but there is no benefit to tuning the weight. We have performed experiments on the CUB dataset with varying the weight ($\lambda\in (0,1]$) of masked auto-encoding objective, i.e., $\mathcal{L}_\texttt{Distill} + \lambda\mathcal{L}_\texttt{MAE}$. As shown in the table below, our proposed self-distillation for further pre-training is insensitive to the weight of the masked auto-encoding objective.
>
> |       Weight for MAE    |  CUB  |
> |:----------------:|:----------------:|
> | $\lambda=1.0$  | $58.06\pm0.90$ |
> | $\lambda=0.5$  | $57.76\pm0.17$ |
> | $\lambda=0.3$  | $58.21\pm0.42$ |
> | $\lambda=0.1$  | $57.76\pm0.33$ |
>
>
> ---
>
>
> [**Q3**]. Can you provide the bound of the distance between the initial weights and teacher weights?
> - Thank you for asking a good question. We have already shown this in Theorem 2. Please recall that the teacher at $t\in\mathbb{N}_+$ round of self-distillation used to be a student of the $t-1$ round of the self-distillation. Since Theorem 2 holds for all non-negative integer $t\in\mathbb{N}_0$,
>
>      the distance between the initial weight and the teacher weight  $\lVert w_\mathrm{init} - w_{t-1, T}\rVert_2$ is upper bounded by $\psi(t-1)$ and the upper bound strictly decreases for all $t\in\mathbb{N}_+$.
>
>
> - Specifically, please note that Theorem 2 also shows the bound on the distance between the weight of the initial teacher model without any self-distillation $w_{0,T}$ and the initial pre-trained weight $w_\mathrm{init}$ for $t=1$. We have revised the paper to clarify this point by adding this explanation as **Remark 1** from **Appendix A**.

---

### Official Review · Reviewer_PVpp · 2022-10-30

**Confidence:** 4
**Correctness:** 4
**Technical Novelty And Significance:** 2
**Empirical Novelty And Significance:** 4
**Recommendation:** 6

**Clarity, Quality, Novelty And Reproducibility:**

Clarity: easy to follow.
Quality: Well-developed and presented.
Novelty: Marginal but principled.
Reproduciblity: easy to implement with a code release.

**Strength And Weaknesses:**

Strengths:
+ This is a somewhat interesting but understandable technique as the distillation loss can function as an ensemble regularization as it is also trained with a MAE loss. This kind of things usually work but knowing that it works for an MAE loss is new to me.
+ The method is well-benchmarked and shows consistent improvements on a set of fine-grained classification tasks (and some language tasks that are out of my scope).

Weaknesses:
- I believe the method works well and I understand that using MAE is preferable due to its simplicity. But I still wonder how it compares with SupCon [A] in the direct fine-tuning setting.
- Missing references to a line of highly-related self-distillation methods that works on a single network [B][C][D].

[A] Supervised Contrastive Learning, NeurIPS 2020.
[B] Deeply-supervised knowledge synergy, CVPR 2019
[C] Be your own teacher: Improve the performance of convolutional neural networks via self distillation, ICCV 2019
[D] Contrastive Deep Supervision, ECCV 2022

**Summary Of The Paper:**

The manuscript studies the problem of unsupervised fine-tuning on a small target dataset before the actual supervised fine-tuning on it. The insight is that even unsupervised fine-tuning (MAE here) on a small dataset causes biased representation that hurts generalization. So the proposed recipe is to use another unsupervised fine-tuned network as a feature distillation regularization term. The authors also provide a theoretical justification using a single-layer network. Evaluations on a set of vision and language tasks show the effectiveness. The method is easy to implemented and codes are provided.

**Summary Of The Review:**

A good empirical study to me.

---

> ### Author Response · Authors · 2022-11-09
> **Response to Reviewer PVpp**
>
> [**Q1**]. Please compare SupCon[1] with MAE in the direct fine-tuning setting.
>
> - As suggested, we fine-tune a pre-trained transformer with supervised contrastive loss and cross-entropy loss. Specifically, following the training detail of the paper [1], we fine-tune an encoder of the transformer with the supervised contrastive loss and train a randomly initialized linear classifier with cross-entropy loss.
>
> - In the table below, supervised contrastive loss significantly degrades the performance of pre-trained transformers. We conjecture that the transformer pre-trained with a masked auto-encoding objective might not be compatible with a contrastive loss and thus we may use the same objective for pre-training and further pre-training.
>
> |                     |  X-rays          | Flower         |
> |------------------|-----------------|----------------|
> | Fine-tuning          | $77.15\pm0.52$  | $88.78\pm0.65$|
> | Further Pre-training | $77.79\pm 2.06$ | $88.63\pm0.35$ |
> | SupCon               | $75.30\pm1.03$  | $76.70\pm1.76$ |
> | Self-Distillation    | $\textbf{79.68}\pm\textbf{1.05}$  | $\textbf{90.28}\pm\textbf{0.44}$ |
>
> ---
>
> [**Q2**] Missing references to a line of highly-related self-distillation methods that work on a single network [2,3,4].
> - Thank you for suggesting related works for self-distillation. We have revised the draft to include them [2,3,4] in the related work section.
>
> ## References
> [1] Khosla, Prannay, et al. "Supervised contrastive learning." Advances in Neural Information Processing Systems 33 (2020): 18661-18673.
>
> [2] Sun, Dawei, et al. "Deeply-supervised knowledge synergy." Proceedings of the IEEE/CVF Conference on Computer Vision and Pattern Recognition. 2019.
>
> [3] Zhang, Linfeng, et al. "Be your own teacher: Improve the performance of convolutional neural networks via self distillation." Proceedings of the IEEE/CVF International Conference on Computer Vision. 2019.
>
> [4] Zhang, Linfeng, et al. "Contrastive Deep Supervision." European Conference on Computer Vision. Springer, Cham, 2022.

---

> > ### Comment · Reviewer_PVpp · 2022-11-14
> > **Response**
> >
> > SupCon experiments: Somewhat understandable.
> >
> > My recommendation remains as BA, but I have to point out that the training pipeline is quite long, when compared with references I listed [ABCD]. I guess this is a paper that I would follow and cite, if proven stably reproducible.

---

### Official Review · Reviewer_5qJ3 · 2022-10-31

**Confidence:** 3
**Correctness:** 3
**Technical Novelty And Significance:** 3
**Empirical Novelty And Significance:** 3
**Recommendation:** 6

**Clarity, Quality, Novelty And Reproducibility:**

The paper is well written and in good quality.

As for novelty, the self-distillation technique has been used and proven to be helpful in many previous works. Therefore, I would rather take this paper as a successful application of self-distillation in the specific tasks.

I assume the proposed method can be easily reproduced based on the authors' description and the attached code.

**Strength And Weaknesses:**

Strength:
1. The proposed method is simple yet effective.
2. The authors provide comprehensive theoretical analysis which clearly shows the role of the proposed algorithm.

Weaknesses:
1. I wonder if it is safe to say the improvement of self-distillation is solely from solving the overfitting problem. In fact besides the original self-distillation paper which adopts this method for better generalization, many other works also use it for better performance without considering overfitting. The authors present the gap between training and testing loss in the experiments, which is great. It would be better if the authors can show that with similar training loss/training accuracy the model train with the proposed method has lower testing loss.
2. This experiments in this paper only focuses on MAE-based model. I think the authors may have some discussion about the effect of the proposed method on other types of self-supervised algorithms, for example, the contrastive learning based methods like SimCLR and DINO.
3. Since the proposed method requires two steps of training, I am not sure if this method is efficient enough.


**Summary Of The Paper:**

This paper focuses on the usage of pretrained vision transformers on downstream tasks. The authors point out the weakness of the current further pretraining strategy about overfitting. To solve that problem, they propose a new pipeline which includes a self-distillation process to distill the knowledge from a further-pretrained model to an initial model.

**Summary Of The Review:**

As mentioned above, this paper is overall good in my opinion. Although the novelty is to some extent limited by applying commonly used technique, the authors present insightful analysis and comprehensive experiments.

---

> ### Author Response · Authors · 2022-11-09
> **Response to Reviewer 5qJ3**
>
> [**Q1**] It would be better to show that the model trained with the proposed method achieves similar train loss but lower test loss than the model with further pre-training.
> - As suggested, we show training and test loss in the below table.  Both further pre-training and self-distillation  reach **near zero training loss**, but the self-distillation achieves much **lower test loss** than the further pre-training. This result empirically verifies that self-distillation mitigates the overfitting issue. We have revised to included the table in **Appendix B**.
>
> | | Aircraft | CUB | DTD | Dogs | X-rays | Flower |
> |:------------------------------:|:----------------------:|:----------------------:|:----------------------:|:----------------------:|:----------------------:|:----------------------:|
> | Further Pre-train Train Loss (log) | $-9.1377 \pm 0.0910$ | $-8.5855 \pm 0.0557$ | $-3.7007 \pm 0.3738$ | $-5.4175 \pm 0.1379$ | $-8.1085 \pm 1.9048$ | $-9.7087 \pm 0.3223$ |
> | Further Pre-train Test Loss | $1.4499 \pm 0.0352$ | $2.4423 \pm 0.0268$ | $2.4967 \pm 0.0862$ | $1.9697 \pm 0.0215$ | $2.2683 \pm 0.2225$ | $0.6368 \pm 0.0438$ |
> | Self-Distillation Train Loss (log) | $-9.1790 \pm 0.0868$ | $-8.1677 \pm 0.3151$ | $-4.0844 \pm 0.3099$ | $-5.5035 \pm 0.0664$ | $-6.0863 \pm 0.5092$ | $-7.9847 \pm 1.2804$ |
> | Self-Distillation Test Loss | $\textbf{1.3645} \pm \textbf{0.0096}$ | $\textbf{2.2622} \pm \textbf{0.0418}$ | $\textbf{2.1510} \pm \textbf{0.0639}$ | $\textbf{1.9045} \pm \textbf{0.0183}$ | $\textbf{2.0800} \pm \textbf{0.1723}$ | $\textbf{0.5052} \pm \textbf{0.0360}$ |
>
> ---
>
> [**Q2**] It would be better to discuss the effect of the proposed method on other types of self-supervised algorithms such as SimCLR and DINO.
>
> - Thank you for your suggestion. We have revised to cite the paper DINO [1] as an example for self-supervised learning methods.  As stated in the introduction, we focus on the masked auto-encoding objective for **generality** so that we can apply it to various domains.
>
> - Please recall that masked Auto-encoding objective does not depend on any data augmentations compared to other self supervised learning methods (SimCLR, DINO) requiring data augmentations to construct positive pairs. Due to the requirement of data augmentations, it is challenging to apply such self supervised learning methods to some target tasks (Chest X-rays or text classification) where there is no well-defined data augmentation.
> ---
> [**Q3**] Since the proposed method requires two steps of training, I am not sure if this method is efficient enough.
> - Thank you for pointing it out. In **Appendix B**, we have revised to include extra computation cost required for further pre-training and self-distillation compared to direct fine-tuning.
>
> - However, please recall that improving the efficiency of adapting pre-trained transformers is **not our goal**. We tackle the overfitting of Vision Transformer when it is further pre-trained on a target dataset, and improve its generalization performance.
>
> - Although self-distillation requires extra training costs, it significantly improves the generalization performance of Vision Transformer and RoBERTA, which shows the effectiveness of regularization induced by self-distillation. This improvement cannot be obtained by further pre-training the transformers longer, as shown in our ablation study.
>
> - For future work, in order to improve training efficiency, we will leverage a meta-learning framework to learn an initialization that can be fast adapted to a target dataset at a further pre-training stage.
>
> ## References
>
> [1] Caron, Mathilde, et al. "Emerging properties in self-supervised vision transformers." Proceedings of the IEEE/CVF International Conference on Computer Vision. 2021.

---

### Official Review · Reviewer_yUym · 2022-11-03

**Confidence:** 4
**Correctness:** 4
**Technical Novelty And Significance:** 3
**Empirical Novelty And Significance:** 3
**Recommendation:** 8

**Clarity, Quality, Novelty And Reproducibility:**

The paper is generally well-written. I think the proposed idea is straightforward and novel. The method seems easy to reproduce.


**Strength And Weaknesses:**

Strength
1. The writing is clear and the proposed method seems straightforward and easy to implement.
2. The authors have provided a theoretical analysis of the effect of the self-distillation process.
3. Experiments are thorough and cover both image classification and text classification with transformers. The experiments are repeated multiple runs to test for statistical significance.

Weaknesses
1. The results, although significantly different, are not better by a large margin compared to fine-tuning, especially for image classification (1-2 points different). It would be great if the authors also provide an analysis of how much additional computational cost there is compared to the finetuning and further pre-training method, which could be a useful reference for people that want to use the proposed method.


**Summary Of The Paper:**

This paper proposes a method for finetuning transformers on new datasets. By an additional self-distillation process between the pre-training and finetuning stages, the authors show that transformers can achieve better generalization on downstream datasets. The authors provide a theoretical analysis of the proposed method with a simplified model. Experiments on ten image and text classification datasets show the efficacy of the proposed method.


**Summary Of The Review:**

I like the clear and novel approach. Although the performance boost is not huge, I think the paper provides enough useful insights and analysis to be accepted.

---

> ### Author Response · Authors · 2022-11-09
> **Response to Reviewer yUym**
>
> [**Q1**] It would be great if the authors also provide an analysis of how much additional computational cost there is compared to the fine-tuning and further pre-training method.
>
> - Thank you for the suggestion. We report the wall clock training time of each method with 3090 RTX GPU and Intel(R) Xeon(R) Silver 4210R CPU.  It takes **32 minutes and 18 seconds for fine-tuning**. We need an extra **1 hour 29 minutes 33 seconds** for **further pre-training** and **5 hours 13 minutes 22 seconds** for **self-distillation**.

---

### Author Response · Authors · 2022-11-09
**Summary of the Revision**

We really appreciate all the reviewers for their constructive comments. We have responded to the individual comments from the reviewers below, and believe that we have successfully responded to most of them. We have included the discussions and results of the suggested experiments in the revision. Here we briefly summarize the updates we have made to the revision:

- We have reported wall clock training time of further pre-training and self-distillation methods in Appendix B, as suggested by Reviewer  **yUym**.

- We have separately reported training and test loss to show that self-distillation mitigates overfitting issue in Appendix B , as suggested by Reviewer **5qJ3**.

- We have cited the paper DINO [1] in section 1 and 2, as suggested by Reviewer **5qJ3**.

- We have included discussion of some relevant works [2,3,4] in section 2, as suggested by Reviewer **PVpp**.

- We have included additional baseline, supervised contrastive learning [5] in Appendix B, as suggested by Reviewer **PVpp**.

- We have included experiments varying the weight of the masked auto-encoding objective in Appendix B, as suggested by **RiKW**.

- We have added Remark 1 to emphasize the distance between the weight of a teacher and the initial pre-trained weight in Appendix A, as suggested by **RiKW**.

---


## References
 [1] Caron, Mathilde, et al. "Emerging properties in self-supervised vision transformers." Proceedings of the IEEE/CVF International Conference on Computer Vision. 2021.

[2] Sun, Dawei, et al. "Deeply-supervised knowledge synergy." Proceedings of the IEEE/CVF Conference on Computer Vision and Pattern Recognition. 2019.

[3] Zhang, Linfeng, et al. "Be your own teacher: Improve the performance of convolutional neural networks via self distillation." Proceedings of the IEEE/CVF International Conference on Computer Vision. 2019.

[4] Zhang, Linfeng, et al. "Contrastive Deep Supervision." European Conference on Computer Vision. Springer, Cham, 2022.

[5]  Khosla, Prannay, et al. "Supervised contrastive learning." Advances in Neural Information Processing Systems 33 (2020): 18661-18673.

---

### Decision · Program_Chairs · 2023-01-20

**Decision:**

Accept: poster

**Justification For Why Not Higher Score:**

Low novelty.

**Justification For Why Not Lower Score:**

Reviewers felt while novelty was low, results are still interesting and useful.

**Metareview: Summary, Strengths And Weaknesses:**

Paper Summary:
Authors propose self-distillation for pre-training before finetuning. A model is pretrained on unlabeled data using self-supervision, then used to self-distill in a repeated self-supervision, before finally being fine-tuned to a target task. Performance is improved compared to other approaches.

Review Summary:

Pros:
- Writing is clear (yUym,RiKW, 7okZ)
- Method is easy to implement (yUym, 5qJ3)
- Seems generalizable to other tasks (7okZ)
- Experiments are thorough and include statistical significance (yUym, PVpp, RiKW,7okZ)
- Good theoretical analysis as to why this works (5qJ3,7okZ)
- Interesting and makes sense why it works because it is essentially an ensemble regularization (PVpp)

Cons:
- Limited novelty but the approach is still valuable (RiKW)
- Difference isn't large. Would be good to better understand additional cost for this training (yUym, 5qJ3) -- authors provided more information.
- Only focused on MAE self-supervision (5qJ3) -- Authors have cited other self supervised methods but not added any new experiments.
- Needs comparison to SupCon (PVpp) -- authors added this data



AC Recommendation: Accept. All reviewers lean acceptance.

**Note From Pc:**

if the above contains the word "oral" or "spotlight" please see: "oral" presentation means -> notable-top-5% and "spotlight" means -> notable-top-25%. As stated in our emails, we are disassociating presentation type from AC recommendations